# Equilibrium Policy Generalization: A Reinforcement Learning Framework for Cross-Graph Zero-Shot Generalization in Pursuit-Evasion Games

**Runyu Lu**[1,2], **Peng Zhang**[3], **Ruochuan Shi**[2,1], **Yuanheng Zhu**[2,1,†],
**Dongbin Zhao**[2,1,†], **Yang Liu**[3], **Dong Wang**[3], **Cesare Alippi**[4,5]
lurunyu17@mails.ucas.ac.cn
njzhangpeng@mail.dlut.edu.cn
{shiruochuan2025,yuanheng.zhu,dongbin.zhao}@ia.ac.cn
{ly,wdice}@dlut.edu.cn
cesare.alippi@usi.ch
[1]School of Artificial Intelligence, University of Chinese Academy of Sciences[*]
[2]Institute of Automation, Chinese Academy of Sciences [3]Dalian University of Technology
[4]The Swiss AI Lab IDSIA, Università della Svizzera italiana [5]Politecnico di Milano
[†]*Corresponding Authors*

## Abstract

Equilibrium learning in adversarial games is an important topic widely examined in the fields of game theory and reinforcement learning (RL). Pursuit-evasion game (PEG), as an important class of real-world games from the fields of robotics and security, requires exponential time to be accurately solved. When the underlying graph structure varies, even the state-of-the-art RL methods require recomputation or at least fine-tuning, which can be time-consuming and impair real-time applicability. This paper proposes an Equilibrium Policy Generalization (EPG) framework to effectively learn a generalized policy with robust cross-graph zero-shot performance. In the context of PEGs, our framework is generally applicable to both pursuer and evader sides in both no-exit and multi-exit scenarios. These two generalizability properties, to our knowledge, are the first to appear in this domain. The core idea of the EPG framework is to train an RL policy across different graph structures against the equilibrium policy for each single graph. To construct an equilibrium oracle for single-graph policies, we present a dynamic programming (DP) algorithm that provably generates pure-strategy Nash equilibrium with near-optimal time complexity. To guarantee scalability with respect to pursuer number, we further extend DP and RL by designing a grouping mechanism and a sequence model for joint policy decomposition, respectively. Experimental results show that, using equilibrium guidance and a distance feature proposed for cross-graph PEG training, the EPG framework guarantees desirable zero-shot performance in various unseen real-world graphs. Besides, when trained under an equilibrium heuristic proposed for the graphs with exits, our generalized pursuer policy can even match the performance of the fine-tuned policies from the state-of-the-art PEG methods.

---

[*]This work was supported in part by the National Natural Science Foundation of China under Grants 62293541, 62293542, and 62476044; in part by the Beijing Natural Science Foundation under Grant 4232056; in part by the Beijing Nova Program under Grant 20240484514; and in part by the International Partnership Program of Chinese Academy of Sciences under Grant 104GJHZ2022013GC.

39th Conference on Neural Information Processing Systems (NeurIPS 2025).

# 1 Introduction

Real-world environments are variable, and the dynamics for real-world games can be time-evolving. As an important class of real-world games, pursuit-evasion games (PEGs) can model a variety of real-world problems (e.g., in robotics and security domains [30; 31; 8]). Typically with a team of pursuers and an adversarial evader, PEGs can be formulated with complex graph structures, where nodes and edges can be temporarily removed or added as the game proceeds. An intelligent game agent should be robust to such changes. In other words, an ideal pursuer or evader policy should be robust to different map or graph inputs. Classical differential game [23; 38] and dynamic programming [32; 14] methods, while accurate, require recomputation under potential changes to game dynamics. In view of this gap, recent works on graph-based PEGs [17; 18; 34; 35] employ deep reinforcement learning (RL) to obtain a more flexible policy. Among the state-of-the-art methods, Grasper (see [17]) can pretrain a generalized multi-agent pursuer policy robust to different initial conditions (e.g., exit positions). Given an actual game situation, subsequent fine-tuning through game-theoretic approaches like PSRO [16] can be used to improve the policy without recomputing from scratch.

However, the state-of-the-art methods still exhibit two practical limitations. First, real-time applicability remains questionable due to the requirement of computationally intensive policy fine-tuning. Recent work [40] points out that existing methods may struggle to adapt to rapid changes in urban settings (e.g., traffic jams, emergencies, and unexpected social events). To make it worse, when the graph structure significantly changes, the pretrained policy will lose the guarantee to be a good starting point for subsequent tuning. Second, existing training processes heavily rely on certain behavior patterns of the opponent policy, which impairs the robustness of the methods. For example, Grasper requires the evader to consistently follow the shortest path to the chosen exits. As a result, the method is not applicable to no-exit PEGs and not general enough for both sides of the players. Besides, such simplifications can make the RL policy more exploitable by an intelligent opponent.

For the first problem, we expect the trained policy to exhibit desirable *zero-shot* performance across different graph structures. For the second problem, we expect the training process to be generally applicable and take strong adversaries into account. Therefore, we ask the following question:

*Is there a general RL framework that can train generalized PEG policies with robust zero-shot performance in unseen graph structures?*

This paper provides a positive answer to this question through the following major contributions:

- We propose Equilibrium Policy Generalization (EPG), a novel reinforcement learning framework that enables zero-shot generalization in Markov PEGs across different graph structures. For the first time, we show that reinforcement learning on a corpus of graphs with equilibrium policies serving as adversaries (and also guidance) can lead to a generalized pursuer (or evader) policy with robust zero-shot performance in unseen real-world graphs.

- For no-exit graphs, we present a dynamic programming (DP) algorithm to efficiently construct an oracle that generates the single-graph equilibrium policies in Markov PEGs. We prove that the proposed algorithm can compute pure-strategy Nash equilibrium under a near-optimal $\tilde{\mathcal{O}}(|S|)$ time complexity. We further extend this algorithm with a grouping mechanism for scalability with respect to pursuer number.

- We design a decentralized network architecture with a novel shortest path distance feature as state inputs to represent cross-graph multi-agent PEG policies. Through ablation studies, we verify that the distance feature, along with our reinforcement learning loss, plays an important role in cross-graph EPG training. Source code can be found in our supplementary material (with DP implementation and test files) and at https://github.com/Cahemgco/EPG_code.

- For graphs with exits, we provide a heuristic approach based on bipartite graph matching as an effective substitute for an exact equilibrium oracle. Directly utilizing this heuristic and approximate equilibrium oracle, we verify that the EPG framework still guarantees a strong zero-shot performance that matches or outperforms the fine-tuned results from the state-of-the-art methods like Grasper [17].

## 2 Preliminaries

### 2.1 Markov Game and Nash Equilibrium

To formulate PEG solving, we introduce two-player zero-sum Markov games and Nash equilibrium.

**Two-player zero-sum Markov game.** An infinite-horizon two-player zero-sum Markov game is represented by a tuple $(S, \mathcal{A}, \mathcal{B}, \mathcal{P}, r, \gamma)$, where $S$ is the state space, $\mathcal{A}$ is the action space of the max-player (i.e., the team of pursuers in a PEG) who aims to maximize the cumulative reward, $\mathcal{B}$ is the action space of the min-player (i.e., the evader in a PEG) who aims to minimize the cumulative reward, $\mathcal{P} \in [0,1]^{|S||\mathcal{A}||\mathcal{B}| \times |S|}$ is the transition probability matrix, $r \in [0,1]^{|S||\mathcal{A}||\mathcal{B}|}$ is the reward vector, and $\gamma \in (0,1)$ is the discount factor.

**Policy and value function.** Following common notations, we denote by $(\mu, \nu)$ the joint policy of the two players, where $\mu$ is the policy of the max-player (pursuers) and $\nu$ is the policy of the min-player (evader). $\mu(s) \in \Delta(\mathcal{A})$ (resp., $\nu(s) \in \Delta(\mathcal{B})$) is the max-player's (resp., min-player's) action distribution at state $s \in S$, and $\mu(s, a)$ (resp., $\nu(s, b)$) is the probability of selecting action $a \in \mathcal{A}$ (resp., $b \in \mathcal{B}$). Define value functions $V^{\mu,\nu}(s) = \mathbb{E}\left[\sum_{t=0}^{\infty} \gamma^t r(s_t, a_t, b_t) \,|\, s_0 = s; \mu, \nu\right]$ and $Q^{\mu,\nu}(s, a, b) = \mathbb{E}\left[\sum_{t=0}^{\infty} \gamma^t r(s_t, a_t, b_t) \,|\, s_0 = s, a_0 = a, b_0 = b; \mu, \nu\right]$ as in single-agent MDPs.

**Nash equilibrium.** A Nash equilibrium (NE) in a game is a joint policy where each individual player cannot benefit from unilaterally deviating from his/her own policy. Specifically, in a two-player zero-sum MG, an NE $(\mu^*, \nu^*)$ satisfies $V^{\mu,\nu^*} \leq V^{\mu^*,\nu^*} \leq V^{\mu^*,\nu}$ for any $\mu$ and $\nu$ at all states. As is well known, every MG with finite states and actions has at least one NE, and all NEs in a two-player zero-sum MG share the same Nash value $V^*(s) = V^{\mu^*,\nu^*}(s) = \max_\mu \min_\nu V^{\mu,\nu}(s) = \min_\nu \max_\mu V^{\mu,\nu}(s)$ [26]. In two-player zero-sum games, Nash equilibrium can be viewed as the optimal joint policy since both players cannot be exploited by their worst-case opponents. Besides, if $(\mu_1, \nu_1)$ and $(\mu_2, \nu_2)$ are both NEs, then $(\mu_1, \nu_2)$ and $(\mu_2, \nu_1)$ are NEs as well [25]. To guarantee the optimality, it suffices to find an equilibrium policy for each player.

### 2.2 Graph-Based Pursuit-Evasion Game

In the context of pursuit-evasion games (PEGs), the transition is usually considered as deterministic, with $\mathcal{P} \in \{0,1\}^{|S||\mathcal{A}||\mathcal{B}| \times |S|}$. As we have mentioned, the max-player is the team of multi-agent pursuers, and the min-player is the single-agent evader. A general target in PEGs is to approximate the Nash equilibrium under the formulation of two-player zero-sum games.

To formulate large-scale problems, a PEG can be represented with a graph $G = \langle \mathcal{V}, E \rangle$, where $\mathcal{V}$ is the set of nodes, and $E$ is the set of undirected edges (which could indicate streets in an urban scenario). When there are $m$ pursuers, the state $s$ can be represented by the locations of all $m + 1$ agents, i.e., $s = (s_p, s_e)$, where $s_p = (v_p^1, v_p^2, \cdots, v_p^m) \in \mathcal{V}^m$, and $s_e = v_e \in \mathcal{V}$. The valid actions for each agent are to move to any node within the current node's neighborhood (including itself). To describe the states where the pursuit has been successful, define a termination function $f : \mathcal{V}^m \times \mathcal{V} \to \{0, 1\}$. When $f(s_p, s_e) = 1$, the game is terminated, and a reward of $+1$ is received. Note that the discount factor $\gamma < 1$ encourages the pursuers to capture the evader as soon as possible in the no-exit scenario. If there are exits in the graph, then another termination function $g : \mathcal{V} \to \{0, 1\}$ should be defined. When $g(s_e) = 1$, the game is terminated as well, but a reward of $-1$ will be received.

## 3 Equilibrium Policy Generalization

As we have mentioned, existing PEG methods can have difficulty adapting to the changes of graph structures or extending to different scenarios. Here, we propose a general reinforcement learning framework called Equilibrium Policy Generalization (EPG) that facilitates cross-graph zero-shot generalization for diverse PEG settings. The overall training pipeline is shown in Figure 1.

Intuitively, a training set (right) consisting of a variety of graphs with different structures is first generated. For each graph $G_i$, an equilibrium policy $(\mu_i^*, \nu_i^*)$ is derived from an equilibrium oracle. Then, reinforcement learning (left) is conducted to learn a generalized policy, with the dynamics constructed through each graph $G$ and opponent $\nu^*$ in the training set. Section 3.1 introduces a detailed construction of the reinforcement learning process established upon soft actor-critic (SAC).

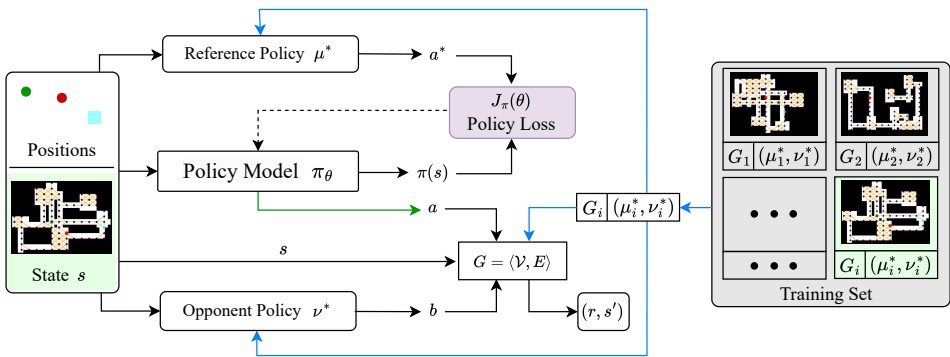

Figure 1: Training pipeline of Equilibrium Policy Generalization

Section 3.2 presents a theoretically sound equilibrium oracle based on dynamic programming (DP). Section 3.3 designs a multi-agent pursuer policy representation specified for cross-graph PEG tasks.

## 3.1 Reinforcement Learning with Equilibrium Adversary

SAC [12; 6] is a well-known off-policy reinforcement learning framework that maximizes expected return regularized under a policy entropy term. Specifically, the value function in discrete-action SAC [6] is defined as $V(s) = \mathbb{E}_{a \sim \pi(s)} [Q(s,a) - \alpha \log \pi(s,a)]$. The losses of the value network $Q_\phi$ and the policy network $\pi_\theta$ are computed as $J_Q(\phi) = \mathbb{E}_{s,a} \left[ \frac{1}{2} (Q_\phi(s,a) - (r + \gamma \mathbb{E}_{s'} [V(s')]))^2 \right]$ and $J_\pi(\theta) = \mathbb{E}_{s,a \sim \pi_\theta(s)} [\alpha \log \pi_\theta(s,a) - Q(s,a)]$, respectively. Besides, the temperature $\alpha$ under target entropy $\overline{H}$ is adaptively updated under loss $J(\alpha) = \mathbb{E}_{s,a} \left[ -\alpha \left( \log \pi(s,a) + \overline{H} \right) \right]$. Here, we use SAC as the foundation algorithm to introduce the RL pipeline in Equilibrium Policy Generalization.

In order to train a generalized policy, our reinforcement learning process goes through the different PEG graphs in the training set. The major difference between our method and the common multi-task learning methods is that we use the equilibrium $\nu^*$ as the opponent policy to construct an adversarial environment for each graph. As is shown in Figure 1, the equilibrium opponent offers an action $b$ that directly serves to generate the transition $(s, a, b, r, s')$ in the current PEG graph $G$. To our knowledge, this is a novel approach not yet considered in the game RL domain. Because $\nu^*$ is hardly exploitable by any strategy, reinforcement learning against such a policy avoids overly exploiting a specific behavior pattern and makes the training process generally focus on cross-graph policy robustness. Since a truly robust policy should not perform poorly against the corresponding equilibrium opponent in any of the graphs, we can use this necessary condition to exclude the weak strategies from the entire policy space. Cross-graph reinforcement learning can be regarded as an efficient way of fulfilling such exclusions. When the training set is diverse enough, only a small area in the parameter space will be left after RL training, where we expect to derive a policy with robust zero-shot generalization.

However, since the equilibrium opponent is hardly exploited and the PEG is a sparse reward environment, the original SAC algorithm can suffer from inefficient exploration even under a single graph. To deal with this problem, the equilibrium $\mu^*$ can also be used as a policy guidance for the training process. Specifically, we regard $\mu^*$ as a reference policy and append an additional KL-divergence term $D_{\mathrm{KL}} (\mu^*, \pi)$ to the original policy loss. When $\mu^*$ is a pure strategy, it holds that $D_{\mathrm{KL}} (\mu^*(s), \pi(s)) = -\log \pi(s, a^*)$, where $a^*$ is the deterministic action of $\mu^*(s)$. In this case, the divergence-regularized policy loss can be computed as $J_\pi(\theta) = \mathbb{E}_{s,a \sim \pi_\theta(s)} [-\beta \log \pi_\theta(s, a^*) + \alpha \log \pi_\theta(s, a) - Q(s, a)]$, where $\beta$ is a hyperparameter that keeps a balance between policy guidance and reinforcement learning.

As is shown in Figure 1, a global state $s$ is sampled under a randomly selected graph $G$ from the training set. The reference policy $\mu^*$ and the model $\pi_\theta$ generate a deterministic action $a^*$ and a strategy $\pi(s)$, respectively. The divergence-regularized policy loss $J_\pi(\theta)$ is then computed, and $\pi_\theta$ is updated through gradient descent. For transition generation, a joint action $(a, b)$ under $s$ is sampled and sent into the graph to compute the reward $r$ and the subsequent state $s'$. In the training phase, $a$ is sampled randomly, and $(r, s')$ is used to train the value network $Q_\phi$ under $J_Q(\phi)$. In the testing phase, $a$ is sampled from the learned policy $\pi_\theta$, and $(r, s')$ is used for trajectory generation and evaluation.

## 3.2 Single-Graph Equilibrium Oracle Construction

As our EPG framework requires the equilibrium policy $(\mu^*, \nu^*)$ for each graph $G$ in the training set, equilibrium oracles should be provided to guarantee applicability in PEGs. In this section, we present effective methods to construct accurate or approximate equilibrium oracles for different scenarios.

### 3.2.1 Dynamic Programming for No-Exit Markov PEGs

Determining whether the pursuers can always capture the evader in a no-exit PEG is proved to be an **EXPTIME-complete** problem (see [11]). This result suggests that an accurate equilibrium oracle could incur a time complexity of $\Omega(|S|)$, where $|S|$ is the number of all game states, which is exponential in the agent number $m + 1$. For sequential PEGs under no-exit graphs, [32] proposes an efficient method to compute the minimum steps for the pursuers to capture the evader. The idea is to iteratively expand the set of states where the pursuit is guaranteed to be successful under the optimal strategy. During the $\mathcal{O}(|S|)$ state expansion, the equilibrium policy is also generated. For Markov games, however, the method is not applicable since the moves for both players are simultaneous.

On the other hand, the classical marking algorithm (see [8]) can have a worst-case time complexity of $\mathcal{O}(|S|^2)$ in practical Markov PEGs. Even when the pursuer number is small (e.g., two or three), the algorithm cannot scale with the number of nodes in the graph, which can be large in the real world. In view of this gap, we first show how to apply the idea of state expansion to simultaneous games and provide a more efficient dynamic programming (DP) algorithm for Markov PEGs.

---

**Algorithm 1:** Dynamic programming (DP) for Markov PEGs in no-exit graphs

**Input:** graph $G = \langle \mathcal{V}, E \rangle$, pursuer number $m$, and termination function $f : \mathcal{V}^m \times \mathcal{V} \to \{0, 1\}$

1   Initialize an empty queue $\mathcal{Q}$ and an array $D = \infty$
2   **for** *pursuer state (positions)* $s_p \in \mathcal{V}^m$ **do**
3     **for** *evader state* $s_e \in \mathcal{V}$ **do**
4       **if** $f(s_p, s_e) = 1$ **then**
5         $D(s_p, s_e) \leftarrow 0$
6         Push $(s_p, s_e)$ into $\mathcal{Q}$
7       **end**
8     **end**
9   **end**
10   **while** $\mathcal{Q}$ *is not empty* **do**
11     Pop the first element $(s_p, s_e)$ from $\mathcal{Q}$
12     **for** *evader neighbor* $n_e \in \text{Neighbor}(s_e), \nexists n'_e \in \mathcal{V}, (n_e, n'_e) \in E, D(s_p, n'_e) > D(s_p, s_e)$ **do**
13       **for** *pursuer neighbor* $n_p \in \text{Neighbor}(s_p) \subset \mathcal{V}^m, D(n_p, n_e) = \infty$ **do**
14         $D(n_p, n_e) \leftarrow D(s_p, s_e) + 1$
15         Push $(n_p, n_e)$ into $\mathcal{Q}$
16       **end**
17     **end**
18   **end**
19   **for** *global state* $(s_p, s_e) \in \mathcal{V}^m \times \mathcal{V}$ **do**
20     $\mu(s_p, s_e) \leftarrow \underset{\text{neighbor } n_p \text{ of } s_p}{\arg\min} \left\{ \underset{\text{neighbor } n_e \text{ of } s_e}{\max} D(n_p, n_e) \right\}$
21     $\nu(s_p, s_e) \leftarrow \underset{\text{neighbor } n_e \text{ of } s_e}{\arg\max} \left\{ \underset{\text{neighbor } n_p \text{ of } s_p}{\min} D(n_p, n_e) \right\}$
22   **end**

**Output:** pursuer policy $\mu : \mathcal{V}^m \times \mathcal{V} \to \mathcal{V}^m$ and evader policy $\nu : \mathcal{V}^m \times \mathcal{V} \to \mathcal{V}$

---

The proposed DP algorithm is shown in Algorithm 1. Intuitively, $D(s_p, s_e)$ is the minimum number of pursuit steps under the optimal pure strategy and is computed through a new form of state expansion (lines 10-18) using a queue $\mathcal{Q}$. Based on $D(\cdot)$, a joint policy $(\mu, \nu)$ is generated through minimax computation (lines 19-22). When a pure-strategy Nash equilibrium exists, the following theorem shows that the computed $D(\cdot)$ induces the Nash value, and the joint policy must be an exact Nash equilibrium. That is to say, the DP algorithm can generate the optimal pure strategies for both players.

**Theorem 1** (Near-optimality of DP policy). *If there exists a pure-strategy Nash equilibrium in the no-exit PEG, then the DP algorithm induces the Nash value $V^*(s = (s_p, s_e)) = \gamma^{D(s_p, s_e)}$ and the corresponding Nash equilibrium $(\mu^*, \nu^*) = (\mu, \nu)$.*

The detailed proof of Theorem 1 is based on mathematical induction and reserved in Appendix A.1. Actually, Algorithm 1 efficiently solves the Bellman minimax equation (see [39]) under the existence of a pure-strategy Nash equilibrium. Note that the update condition $D(s_p, s_e) = \infty$ guarantees that every state $(s_p, s_e)$ is pushed into and popped from $\mathcal{Q}$ at most once. Therefore, the time complexity of Algorithm 1 is a near-optimal $\tilde{\mathcal{O}}(|S|)$, where $\tilde{\mathcal{O}}$ hides the logarithm factors required for picking each $(n_p, n_e)$ through preserving data structures like balanced trees. Also note that the computation of $\mu(s)$ and $\nu(s)$ can be reserved online and thus does not affect the overall time complexity.

Algorithm 1 suggests that the Nash value admits an efficient estimation when the PEG only admits a one-sided termination function. Compared to direct value iteration (see [14]), the DP algorithm can exactly compute the equilibrium policy within finite iterations. The existence of pure-strategy Nash equilibrium implies that a successful pursuit is guaranteed for all $s \in S$ (i.e., $D(s) < \infty$). For example, the optimal strategy in any tree-form $G$ is a pure-strategy Nash equilibrium if we regard adjacency as the condition of a successful pursuit. Even when this assumption does not globally hold in a PEG, Algorithm 1 still induces a near-optimal pursuit strategy for the states with finite $D(\cdot)$.

### 3.2.2 Heuristic Approach for Multi-Exit PEGs and Grouping Extension of DP Approach

For PEGs with exits, it can be more difficult to design a DP-like equilibrium oracle with rigorous guarantees due to the existence of the other termination function $g$. Nevertheless, we observe that the cooperative behaviors among pursuers can be approximately abstracted as one-to-one exit allocation. Based on this observation, we provide an equilibrium heuristic featuring bipartite graph matching to approximately generate equilibrium policies. The detailed construction is reserved in Appendix B due to space limitations. Compared to the DP approach in no-exit scenarios, the heuristic approach computes the current policy for each state independently and can be directly executed during RL training, with a time complexity polynomial in the pursuer number for any current state. Note that the polynomial time guarantee also implies its scalability with respect to agent number.

In order to facilitate the scalability of the DP approach in no-exit graphs with many pursuers, here we further extend DP with a grouping mechanism to trade optimality for applicability. Note that the DP algorithm is directly applicable when there are two or three pursuers. For a large pursuer number $m$, we can express it as the summation of twos and threes. For example, if there are $m = 6$ pursuers, we have that $6 = 2 + 2 + 2$ and can thus group them into three sub-teams, each with two pursuers. For the pursuer side, we can simply use the exact 2-pursuer policies under an arbitrary grouping result to construct a 6-pursuer policy, which is empirically strong due to the optimality of the DP algorithm.

For the evader side, we expect that it should not be easily exploited by the grouping-based pursuers. However, it does not know the exact result of grouping. Therefore, we follow minimax criteria to construct the evader policy at any current state $s$. We use $s_g = (s_g^1 = (s_p^1, s_e), \cdots, s_g^k = (s_p^k, s_e)) \in S_g$ to denote the in-team states for the global state $s$ under any possible grouping result with $k$ sub-teams. Then, we compute $s_* = \arg\min_{s_g \in S_g} \{\max_{i=1}^k D(s_g^i)\}$, where $D(s_g^i) = D(s_p^i, s_e)$ is the computed result from the DP algorithm. The evader policy is to follow the DP policy $\nu^*(s_*^j)$, where $j = \arg\max_{i=1}^k D(s_*^i)$. Intuitively, the DP-based evader always considers the worst-case grouping and ensures that it is hard for at least one sub-team to capture it in this case. While introducing an extra online time complexity, the grouping mechanism avoids the exponentially growing complexity of running Algorithm 1 and makes the DP approach applicable to the scenarios with many pursuers.

### 3.3 Cross-Graph Representation of PEG Policy

To represent a generalized PEG policy, the network architecture should be capable of encoding the meaningful state information across different graph structures. Besides, since the policy should be applicable to the team of homogeneous pursuers, it is appealing to design a decentralized architecture with shared parameters. Such a network architecture can be robust to the change of agent number and allow for decentralized execution when necessary. Under the principle of sequential decision-making, a joint policy can be decomposed as $\pi(a_1, a_2, \cdots, a_m | s) = \prod_{l=1}^m \pi(a_l | s, a_1, \cdots, a_{l-1})$, where $(s, a_1, \cdots, a_{l-1})$ indicates the global state after the first $l - 1 < m$ pursuers take actions $(a_i)_{i \in [l-1]}$.

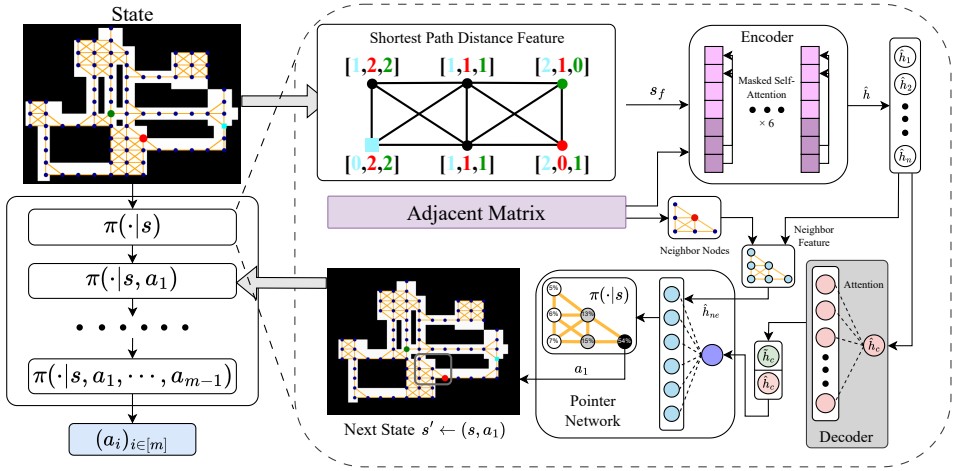

Figure 2: Sequence model with cross-graph policy representation

With the above-mentioned considerations, we present a sequence model with an attention-based architecture (see Figure 2) to represent the cross-graph joint policy. For a team of pursuers with $m$ agents, the sequence model queries the policy network $m$ times under a fixed adjacent matrix $M \in \{0,1\}^{n \times n}$ ($n = |\mathcal{V}|$) for the current graph. The input is composed of a state feature $s_f$ that describes the current global state and the information of node index $c$ for the current acting agent. To uniformly capture the state information across different graphs, we use the **shortest path distances** to the $m + 1$ agents (including the evader) and the exits (if existing) as the initial feature of each node $v \in \mathcal{V}$. The state feature $s_f$ is composed of the normalized features of all $n$ nodes. The shortest path is an important and well-examined concept in graph theory [2], and the distance between any two nodes can be preprocessed using the $\mathcal{O}(n^3)$ Floyd algorithm. In Appendix A.2, we further prove that the input $(s_f, c)$ is dense enough to identify the current global state in sequential decision-making.

Given the state feature input $s_f$, we borrow the ideas from the existing works on robot exploration [3; 19] to construct a policy network (on the right of Figure 2). We first embed $s_f$ into $\mathbb{R}^{d \times n}$ and send it into an encoder composed of multiple self-attention [29] layers, where $d$ is the embedding dimension. Each layer takes the output $h$ of the last layer as the input and outputs $h'$ using a masked attention:

$$q_i = W_Q h_i, k_i = W_K h_i, \ v_i = W_V h_i, u_{ij} = \frac{q_i^T k_j}{\sqrt{d}}, w_{ij} = \frac{e^{u_{ij}}}{\sum_{t=1}^{n} e^{u_{it}}}, h_i' = \sum_{j=1}^{n} \min \{w_{ij}, M_{ij}\} v_j,$$

where $W_Q, W_K, W_V \in \mathbb{R}^{d \times d}$ are the weights to be learned. Theoretically, it requires **Diameter**$(G)$ layers to globally broadcast the local information of a node since $M$ is the adjacent matrix. However, in contrast to using a low-level node feature with indicators (see Grasper [17]), using a high-level distance feature allows the node itself to encode long-term information. With the shortest path distance feature, we can use a fixed number of attention layers (6 in practice) for information transmission.

Denote by $\hat{h}$ the output of the encoder, and recall that $c$ is the node index corresponding to the current acting agent. We further use a decoder without masking to gather global information. Specifically, the decoder uses $\hat{h}_c$ to query in the output features $\hat{h}$ of all nodes, with the keys equal to the values: $q = W_Q \hat{h}_c, k_i = W_K \hat{h}_i, v_i = W_V \hat{h}_i, u_j = \frac{q^T k_j}{\sqrt{d}}, w_j = \frac{e^{u_j}}{\sum_{t=1}^{n} e^{u_t}}, \tilde{h}_c = \sum_{j=1}^{n} w_j v_j$. The decoder output $\tilde{h}_c$ is further concatenated with $\hat{h}_c$ and projected into $\mathbb{R}^d$. Then, it is used as a query for a pointer network [33], which takes the features of the neighbor nodes $\hat{h}_{ne}$ for the current agent as the keys and values. The pointer network directly outputs the attention vector $w$ as the current policy (i.e., $\pi(a|s) = w_a$) since the number of the neighbors aligns with the number of the valid actions.

After the first query through the policy network, an action $a_1$ for the first agent is sampled from $\pi(a|s)$, and the state is updated as $s' = (s, a_1)$. The subsequent queries follow the same process described above. Under the decomposition of sequential decision-making, while the process generates the joint action $(a_i)_{i \in [m]}$ sequentially, it is equivalent to a direct sampling from the joint policy. Note that querying the policy model is practically efficient, especially with the help of GPUs. When the graph structure changes, we simply rerun the Floyd algorithm, instead of the $\tilde{\mathcal{O}}(n^{m+1})$ DP algorithm.

Table 1: Performance of RL pursuer / evader against DP oracle in no-exit PEGs with 2 pursuers

| Graph Structure | Pursuit Success Rate ↑ | | | Evasion Timestep ↑ | |
|---|---|---|---|---|---|
| | DP - DP | $\mathbf{RL}_p$ - DP | SPS - DP | DP - DP | DP - $\mathbf{RL}_e$ |
| Grid Map | 1.00 | 1.00 | 1.00 | $12.29 \pm 2.06$ | $11.88 \pm 2.39$ |
| Scotland-Yard Map | 1.00 | 0.99 | 0.17 | $15.13 \pm 2.77$ | $12.57 \pm 2.96$ |
| Downtown Map | 1.00 | 0.99 | 0.17 | $14.22 \pm 3.27$ | $11.83 \pm 3.12$ |
| Times Square | 1.00 | 0.98 | 0.14 | $16.47 \pm 3.23$ | $14.68 \pm 3.11$ |
| Hollywood Walk of Fame | 1.00 | 0.62 | 0.02 | $25.56 \pm 5.03$ | $20.00 \pm 4.99$ |
| Sagrada Familia | 1.00 | 0.66 | 0.04 | $21.88 \pm 4.82$ | $17.89 \pm 4.59$ |
| The Bund | 1.00 | 0.60 | 0.13 | $25.26 \pm 6.17$ | $20.59 \pm 5.59$ |
| Eiffel Tower | 1.00 | 0.97 | 0.81 | $23.42 \pm 6.48$ | $18.47 \pm 6.12$ |
| Big Ben | 1.00 | 0.91 | 0.13 | $27.89 \pm 6.35$ | $21.58 \pm 6.38$ |
| Sydney Opera House | 1.00 | 0.74 | 0.13 | $26.92 \pm 5.89$ | $22.37 \pm 6.16$ |

## 4 Experiments

In this section, we verify that our EPG framework can train a generalized policy with robust zero-shot performance under unseen graph structures. Using 76 procedurally generated maps in the Dungeon environment [5], we construct a heterogeneous training set by discretizing each map into small-scale (100-node) and large-scale (500-node) graphs. The policy training follows the cross-graph reinforcement learning pipeline in Section 3.1. In no-exit scenarios with 2 or 3 pursuers, we directly use the DP algorithm (Algorithm 1) in Section 3.2 as an accurate equilibrium oracle. When there are more pursuers, we use the grouping extension of the DP approach to construct an approximate equilibrium oracle. In multi-exit scenarios, we use the heuristic approach to construct the approximate oracle. Besides, we use the sequence policy model in Section 3.3 to represent either pursuer or evader policy. For the evader side, the policy model is reduced to a single policy network, with $\mu^*$ and $\nu^*$ exchanged in Figure 1. The training time costs are reported in Appendix C.3. We test the zero-shot performance of the learned pursuer or evader policy under unseen graphs without further fine-tuning.

### 4.1 Performance Tests in Real-World No-Exit Graphs

In no-exit graphs, the DP policy generated by Algorithm 1 is the optimal pure strategy for both players when the pursuit is guaranteed for all states. With this theoretical guarantee, we can use the DP oracle to construct the adversarial opponent and benchmark the zero-shot performance of the RL pursuer or evader. Our test graphs include Grid Map (a $10 \times 10$ grid), Scotland-Yard Map (from the board game Scotland-Yard), Downtown Map (a real-world location from Google Maps), and 7 famous real-world spots (from Times Square to Sydney Opera House). The real-world graph structures for testing are illustrated in Figure 8. We set the termination function $f(s)$ to be 1 when half of the pursuers are simultaneously adjacent to the evader. This moderate success condition guarantees pursuit for all synthetic and real-world graphs described in the main paper when the pursuer number is more than 1.

We first evaluate the zero-shot performance of the trained RL policies against the accurate DP oracle in no-exit PEGs with 2 pursuers. The result under each graph is averaged over 500 tests, each of which is forced to terminate after 128 steps (corresponding to the length of an episode). The pursuit is guaranteed to be successful under the DP pursuer, while the DP evader attempts to avoid capture by maximizing the termination (evasion) timesteps. As is shown in Table 1, when we use the RL pursuer trained through EPG to replace the DP pursuer, the pursuit success rates are still kept over 0.6. In comparison, if we simply use a shortest path strategy (SPS) to directly approach the DP evader, the success rates can be sensitive to graph structures and low on average. On the other hand, when we use the RL evader to replace DP, the evasion timesteps do not drop significantly. The results demonstrate that our RL policies generalize well across graphs against the unexploitable DP opponent policies.

Now we evaluate the zero-shot performance of the trained RL policies against different opponents. Besides the DP opponent, we show the results of our RL pursuer against our RL evader in Table 2. The success rate of RL - RL is always 1 and thus omitted. Furthermore, we include the results of the approximately best-responding (BR) policies directly trained against our RL policies on the test graphs. Ideally, such results could reflect the worst-case performance of our RL policies. However,

Table 2: Performance comparison of RL pursuer / evader against different opponents

| Graph Structure | Pursuit SR ↑ | | Evasion Timestep ↑ | | |
|---|---|---|---|---|---|
| | RL - BR | RL - DP | RL - RL | BR - RL | DP - RL |
| Grid Map | 1.00 | 1.00 | $14.34 \pm 5.08$ | $11.67 \pm 2.60$ | $11.88 \pm 2.39$ |
| Scotland-Yard Map | 1.00 | 0.99 | $17.82 \pm 7.31$ | $13.12 \pm 3.65$ | $12.57 \pm 2.96$ |
| Downtown Map | 0.99 | 0.99 | $17.65 \pm 8.72$ | $13.81 \pm 5.83$ | $11.83 \pm 3.12$ |
| Times Square | 0.95 | 0.98 | $20.29 \pm 9.65$ | $16.20 \pm 5.16$ | $14.68 \pm 3.11$ |
| Walk of Fame | 0.67 | 0.62 | $34.34 \pm 20.18$ | $20.22 \pm 9.60$ | $20.00 \pm 4.99$ |
| Sagrada Familia | 0.76 | 0.66 | $26.87 \pm 11.00$ | $18.83 \pm 6.30$ | $17.89 \pm 4.59$ |
| The Bund | 0.56 | 0.60 | $30.66 \pm 17.27$ | $21.91 \pm 10.58$ | $20.59 \pm 5.59$ |
| Eiffel Tower | 0.98 | 0.97 | $28.41 \pm 16.80$ | $19.28 \pm 10.48$ | $18.47 \pm 6.12$ |
| Big Ben | 0.94 | 0.91 | $33.24 \pm 18.71$ | $23.07 \pm 12.91$ | $21.58 \pm 6.38$ |
| Sydney Opera House | 0.80 | 0.74 | $32.94 \pm 14.72$ | $25.78 \pm 16.79$ | $22.37 \pm 6.16$ |

the BR policies are generally hard to train in practice, possibly because the EPG policies are fairly strong. We record their best results during 20000 training episodes. As is shown in Table 2, while the BR policies are better at exploiting our RL policies in comparison with our RL opponent policies, they show no advantage in comparison with DP. Therefore, considering the results in Table 1, we can say that EPG policies have robust cross-graph zero-shot performance under unseen graph structures.

## 4.2 Ablation Study and Extended Evaluations

Here we analyze the important aspects of our cross-graph policy training. We use 10 unseen Dungeon maps and their corresponding graphs as a testing set. We conduct the ablation study by comparing the termination timesteps of different pursuer policies against the DP evader in no-exit PEGs.

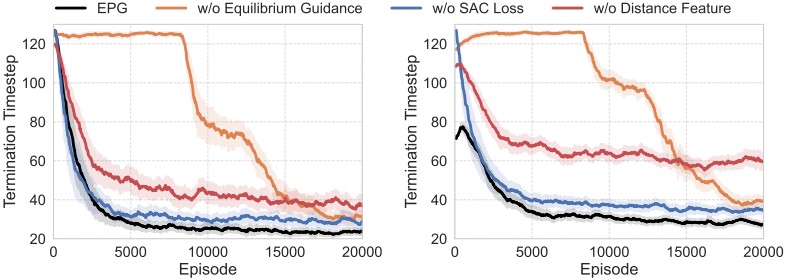

Figure 3: Termination timesteps of pursuer policies under training set (left) and testing set (right)

As is shown in Figure 3 (error bars representing standard deviations), when we remove the equilibrium guidance from our framework, it is less efficient to train the generalized pursuer policy under the remaining SAC loss (orange). This phenomenon verifies that the policy exploration of SAC itself can have low efficiency under the sparse reward environment with an equilibrium adversary. When we remove the original SAC loss, the training becomes a kind of supervised learning, and the learned policy suffers from a clear performance decline due to the lack of the reward signal (blue). Therefore, it is reasonable to combine equilibrium guidance with reinforcement learning against an equilibrium adversary for effective policy generalization (black). If we directly use the 2-dimensional position and the agent indicators to replace the shortest path distances as the feature of each node, the learned policy suffers from a significant performance decline and cannot guarantee its zero-shot performance (red). This verifies that our distance feature is suitable for cross-graph generalization of PEG policies.

We have also conducted some extended evaluations under no-exit PEGs. We consider the scenario with 6 pursuers and use the proposed grouping mechanism to construct an approximate DP oracle. The corresponding results are provided in Table 5, showing even higher success rates of RL pursuit in comparison with Table 1. In Appendix E, we further show that our RL pursuers can have a zero-shot performance even better than DP when the pursuit is not globally guaranteed. Additionally, while the pursuer policy is trained on a graph set with an average node number of $152.24$, we find that it

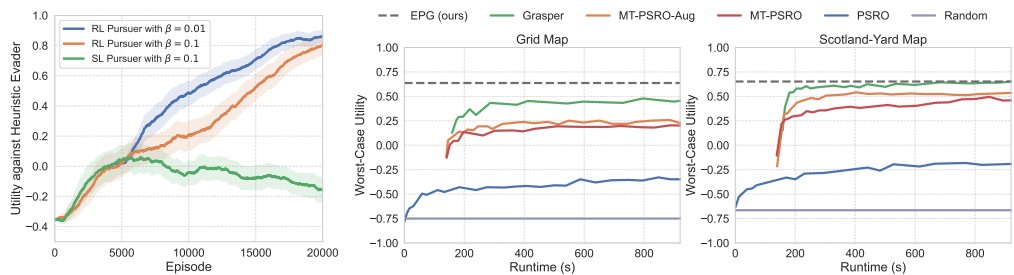

Figure 4: Zero-shot performance in $8$-exit PEGs during training (left) and testing (mid & right)

can directly generalize to large-scale graphs possibly with more than $1000$ nodes (Appendix F.1). Besides, we find that our policies significantly outperform the PSRO policies directly trained on the test graphs (Appendix F.2). These results further demonstrate the versatility of our EPG framework.

### 4.3 Performance Comparisons in Graphs with Exits

Since the state-of-the-art approaches in graph-based PEGs (including Grasper [17] and multi-task PSRO (MT-PSRO) [18]) deal with multi-exit PEGs, we also apply our EPG framework to the same problem setting of $5$ pursuers, $1$ evader, and $8$ exits described in the paper of Grasper [17]. As is mentioned in Section 3.2, we design an equilibrium heuristic based on bipartite graph matching (see Appendix B) to construct an approximate equilibrium oracle. As is shown in Figure 4 (left), our EPG framework steadily trains the RL pursuers under different hyperparameters $\beta \in \{0.01, 0.1\}$ when using the heuristic approach. Mere supervised learning (green) without SAC loss can no longer guarantee a steady improvement in unseen graphs, though it can simply work in the no-exit scenario.

Since [17] also uses Grid Map and Scotland-Yard Map as two fixed testing graphs, it is direct to evaluate the zero-shot performance of our RL pursuers under the same game rules. The comparative methods of Grasper, MT-PSRO, and MT-PSRO with augmentation (MT-PSRO-Aug) all train their policies given the graph structure. Nevertheless, Figure 4 (mid & right; with results from [17]) shows that our zero-shot performance (the dashed lines) under the unseen graphs ($0.637$ for Grid Map and $0.652$ for Scotland-Yard Map) significantly outperforms their zero-shot performance (the starting points), which is with respect to unseen initial conditions. Even after a 10-minute fine-tuning process using PSRO [16], only Grasper in the Scotland-Yard Map can match the zero-shot performance of our policy. This result further demonstrates the generalization capability of our EPG framework.

## 5 Conclusion

This paper proposes Equilibrium Policy Generalization (EPG), a novel reinforcement learning framework for training generalized PEG policies with robust zero-shot performance across graph structures. Established upon the idea of constructing adversary and guidance with equilibrium policies, EPG features a general cross-graph RL pipeline applicable to both pursuer and evader sides in both no-exit and multi-exit scenarios. For no-exit PEGs, we propose a dynamic programming (DP) algorithm as an equilibrium oracle and theoretically prove that it generates pure-strategy Nash equilibrium with near-optimal time complexity. We further extend DP with a grouping mechanism and equip RL with a sequence model to facilitate scalability. Experiments and ablation studies verify that our EPG framework can effectively train a generalized pursuer or evader policy with robust zero-shot performance in unseen real-world graphs. When trained with a matching-based equilibrium heuristic that we propose for multi-exit PEGs, the RL pursuer policy exhibits a strong zero-shot performance that matches or outperforms the fine-tuned results from the state-of-the-art methods.

Although this paper focuses on pursuit-evasion games and thus introduces handcrafted DP/heuristic equilibrium oracles, the EPG framework is in principle applicable to any other game scenario through the use of general and approximate oracles like PSRO. One limitation of this work is that we only focus on the case where both players have perfect information of the game state. It can be interesting to further examine whether the perfect-information equilibrium oracles are still beneficial for robust policy learning under an imperfect-information or partially observable game setting.

## Acknowledgments and Disclosure of Funding

This work was supported in part by the National Natural Science Foundation of China under Grants 62293541, 62293542, and 62476044; in part by the Beijing Natural Science Foundation under Grant 4232056; in part by the Beijing Nova Program under Grant 20240484514; and in part by the International Partnership Program of Chinese Academy of Sciences under Grant 104GJHZ2022013GC. We also thank all the reviewers for providing valuable comments that helped us improve this paper.

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

## A  Omitted Proofs

### A.1  Proof of Theorem 1

*Proof.*  For no-exit PEGs, the Nash value satisfies the following Bellman minimax equation:

$$
V^*(s) = \begin{cases} \max\limits_{\mu(s)\in\Delta(\mathcal{A})} \min\limits_{b\in\mathcal{B}} \sum\limits_{a\in\mathcal{A}} \mu(s,a)\left(r(s,a,b) + \gamma \sum\limits_{s'\in S} \mathcal{P}(s,a,b,s')V^*(s')\right), & f(s)=0 \\ 1, & f(s)=1 \end{cases}
$$

Since the transition is deterministic and a non-zero reward is received only when a termination state is reached, we can simplify the Bellman equation as follows:

$$
V^*(s) = \begin{cases} \max\limits_{\mu(s)\in\Delta(\mathcal{A})} \min\limits_{b\in\mathcal{B}} \sum\limits_{a\in\mathcal{A}} \mu(s,a)\gamma V^*(s' = \mathcal{P}(s,a,b)), & f(s)=0 \\ 1, & f(s)=1 \end{cases}
$$

The equilibrium policy for the max-player satisfies:

$$
\mu^*(s) \in \operatorname*{arg\,max}_{\mu(s)\in\Delta(\mathcal{A})} \left\{ \min_{b\in\mathcal{B}} \sum_{a\in\mathcal{A}} \mu(s,a)V^*(s'=\mathcal{P}(s,a,b)) \right\}
$$

When there is a pure-strategy Nash equilibrium in the game, the $\arg\max$ has a pure-strategy solution, and the Bellman equation can be further simplified:

$$
V^*(s) = \gamma \max_{a\in\mathcal{A}} \min_{b\in\mathcal{B}} V^*(s' = \mathcal{P}(s,a,b)) \tag{1}
$$

Note that the Nash value has the form of $V^*(s) = \gamma^d (d \in \mathbb{N})$. Therefore, we consider using mathematical induction. We assume that $V^*(s) = \gamma^{D(s)}$ holds for all states $s$ that satisfies either $V^*(s) = \gamma^d$ or $D(s) = d$ when $d < k$ ($\gamma^d > \gamma^k$). We want to prove that $V^*(s) = \gamma^{D(s)}$ holds for all states $s$ that satisfies either $V^*(s) = \gamma^k$ or $D(s) = k$. Clearly, our initialization guarantees that the proposition holds for $k = 0$. Our update condition $D(n_p, n_e) = \infty$ guarantees that every state $s \in S$ is pushed into and popped from $\mathcal{Q}$ at most once. Note that the following proof reverses the notations of $s$ and $s'$ in (1) to better align with $s = (s_p, s_e)$ in Algorithm 1.

Now, we prove the first half of the proposition. For an arbitrary state $s' = (n_p, n_e)$ that satisfies $V^*(s') = \gamma^k$, the simplified Bellman equation (1) guarantees that there exists $a = s_p \in \mathcal{A}(n_p)$ and $b = s_e \in \mathcal{B}(n_e)$ such that $V^*(s') = \gamma V^*(s = \mathcal{P}(s',a,b))$. Therefore, there exists $s = (s_p, s_e)$ such that $V^*(s) = \gamma^{k-1}$. According to the first half of the induction hypothesis, we have that $D(s) = k - 1 < \infty$, which implies that the algorithm once pushed $s'$ into $\mathcal{Q}$. Besides, the Bellman equation guarantees that $\forall b' \in \mathcal{B}(n_e), V^*(\mathcal{P}(s',a,b')) \geq V^*(\mathcal{P}(s',a,b)) = V^*(s) = \gamma^{k-1} > \gamma^k$. By induction hypothesis, $D(s_p, n'_e) \leq D(s_p, s_e)$ holds for any neighbor $n'_e$ of $n_e$. Therefore, the algorithm must enumerate $n_e$ when popping $s = (s_p, s_e)$. If we have $D(n_p, n_e) = \infty$ at the moment, then $n_p$ will be enumerated in the inner loop, and we will have $D(n_p, n_e) = D(s_p, s_e) + 1 = k$. Now we complete the proof by showing that $D(n_p, n_e) < \infty$ implies $D(n_p, n_e) = k$. Actually, if $k < D(n_p, n_e) < \infty$, then $D(s')$ must be computed by adding 1 to some $D(s'') \geq k$. Since $s''$ must be popped from $\mathcal{Q}$ no later than $s$, it is contradictory to the fact that $D(s'') > D(s) = k - 1$. If $D(n_p, n_e) < k$, then the second half of the induction hypothesis implies that $V^*(s') = \gamma^{D(n_p,n_e)}$, which is contradictory to the fact that $V^*(s') = \gamma^k$.

Then, we prove the second half of the proposition. For an arbitrary state $s' = (n_p, n_e)$ that satisfies $D(s') = k$, the $D(s)$ must be computed by adding 1 to some $D(s) = k - 1$, where $s = (s_p, s_e)$. According to the first half of the induction hypothesis, we have $V^*(s) = \gamma^k$. The algorithm guarantees that $D(s_p, n'_e) \leq D(s_p, s_e) = k-1$ holds for any neighbor $n'_e$ of $n_e$. By induction hypothesis, it holds that $\forall b' \in \mathcal{B}(n_e), V^*(\mathcal{P}(s',a,b')) \geq V^*(\mathcal{P}(s',a,b))$ when $a = s_p \in \mathcal{A}(n_p)$ and $b = s_e \in \mathcal{B}(n_e)$. Therefore, $\min\limits_{b\in\mathcal{B}(n_e)} V^*(\mathcal{P}(s',a,b)) = \gamma^{k-1}$ when $a = s_p \in \mathcal{A}(n_p)$. If there exists $a^\dagger = s_p^\dagger \in \mathcal{A}(n_p)$ such that $\min\limits_{b\in\mathcal{B}(n_e)} V^*(\mathcal{P}(s',a',b)) > \gamma^{k-1}$, then we let $b^\dagger = \operatorname*{arg\,min}\limits_{b\in\mathcal{B}(n_e)} V^*(\mathcal{P}(s',a^\dagger,b)) > \gamma^{k-1}$

and let $s^\dagger = (s_p^\dagger, s_e^\dagger = b^\dagger)$. According to the first half of the induction hypothesis, $D(s_p^\dagger, n_e') \leq D(s_p^\dagger, s_e^\dagger) < k-1$ holds for any neighbor $n_e'$ of $n_e$. Since $D(s_p^\dagger, s_e^\dagger) < D(s_p, s_e)$, $s^\dagger$ must be popped from $\mathcal{Q}$ earlier than $s$, which means that $D(s) = \infty$ when $s^\dagger$ is popped. Therefore, $s' = (n_p, n_e)$ must be enumerated when $s^\dagger$ is popped, which is contradictory to the fact that $D(s') = \infty$ when $s$ is popped. Therefore, $V^*(s') = \gamma \max_{a \in \mathcal{A}} \min_{b \in \mathcal{B}} V^*(\mathcal{P}(s', a, b)) = \gamma^k$.

For now, we have proved that $V^*(s) = \gamma^{D(s)}$. Therefore:

$$\mu(s_p, s_e) = \underset{\text{neighbor } n_p \text{ of } s_p}{\arg\min} \left\{ \max_{\text{neighbor } n_e \text{ of } s_e} D(n_p, n_e) \right\} \Rightarrow \mu(s) = \underset{a \in \mathcal{A}}{\arg\max} \min_{b \in \mathcal{B}} V^*(\mathcal{P}(s, a, b))$$

$$\nu(s_p, s_e) = \underset{\text{neighbor } n_e \text{ of } s_e}{\arg\max} \left\{ \min_{\text{neighbor } n_p \text{ of } s_p} D(n_p, n_e) \right\} \Rightarrow \nu(s) = \underset{b \in \mathcal{B}}{\arg\min} \max_{a \in \mathcal{A}} V^*(\mathcal{P}(s, a, b))$$

As there exists a pure-strategy Nash equilibrium $(\mu^*, \nu^*)$, it is directly guaranteed that $(\mu, \nu)$ is a Nash equilibrium. $\qquad\square$

### A.2 Proof of Proposition 1

**Proposition 1** (Basic property of the shortest path distance feature). *When the game is collision-free with respect to the agents' node indexes, the input $(s_f, c)$ is sufficient to reconstruct the global state with the current agent order $l$ for sequential decision-making.*

*Proof.* Recall that the global state $(s_p, s_e) = s_p = (v_p^1, v_p^2, \cdots, v_p^m, v_e)$, where $m$ is the number of pursuers.

Note that $s_f \in \mathbb{R}^{(m+1) \times n}$ contains the normalized shortest path distances of all $n = |\mathcal{V}|$ nodes to all $m+1$ agents. Since the distance is zero only when the agent is at the node, $s_f(k, v) = 0$ implies $v = v_p^k$ when $k \leq m$ and implies $v = v_e$ when $k = m + 1$. Therefore, the global state $(s_p, s_e)$ can be reconstructed by checking the zeros in $s_f$.

When the game is collision-free with respect to the agents' node indexes, there is at most one zero in each column of $s_f$. Since $c$ is the index of the current pursuer agent, there must be exactly one zero in the $c$-th column. Let $v_c$ be the $c$-th node in the graph. Since $s_f(l, v_c) = 0$, the current agent order $l$ can also be obtained by checking the zero in the $c$-th column of $s_f$. $\qquad\square$

# B Equilibrium Heuristic for PEGs with Exits

In no-exit graphs, the termination function $f(s_p, s_e)$ is 1 when a number of pursuers are adjacent to the evader. Since there is no termination condition about exits, the pursuers only care about how soon they can capture the evader. In graphs with exits, however, the pursuers lose when $g(s_p, s_e) = 1$, and a successful pursuit requires strict overlapping rather than adjacency (see [17]). Therefore, to block the exits around the evader is more reasonable than to block all possible actions of the evader and capture it. For the evader, it is in turn reasonable to select an exit that cannot be blocked in time.

In the multi-exit scenario, a single pursuer is enough to block a single exit if it is at least as close as the evader with respect to the shortest path distance. Also, it is theoretically impossible for multiple pursuers to block the exit if none of them is as close, since the evader can simply follow the shortest path towards the exit. Actually, if there exists one pursuer who can block the evader on its shortest path in time, it directly implies that the pursuer has a path to the exit no longer than the evader's, which contradicts the premise. Therefore, the cooperative behaviors among pursuers can be approximately abstracted as one-to-one exit allocation, which can be further formulated by bipartite graph matching.

With the above-mentioned considerations, we construct an equilibrium heuristic based on bipartite graph matching to efficiently generate reasonable policies for both players in multi-exit PEGs. Given the current state of the game, the heuristic algorithm can be described by the following four steps:

- Under the current state $s$, construct a bipartite graph $G_b = \langle (\mathcal{V}_{exit}, \mathcal{V}_{pursuer}), E_b \rangle$, where the nodes in $\mathcal{V}_{exit}$ correspond to the exits in the PEG graph $G$ and the nodes in $\mathcal{V}_{pursuer}$ correspond to the pursuers. There is an edge $e_b \in E_b$ between an exit and a pursuer if and only if the pursuer's shortest path distance to the exit is not longer than the evader's. Intuitively, the existence of an exit-pursuer edge means that the pursuer can reach the exit no later than the evader and thus block the exit in time.

- If a pursuer node has no related edges, which means it cannot block any exit, then it is removed from $G_b$, and this pursuer's current policy is to follow the shortest path towards the evader. If an exit node has no related edges, which means it cannot be blocked by any pursuer, then it is also removed from $G_b$. For the remaining exit nodes, sort them by their distance to the evader in an ascending order.

- Compute the maximum $k$ that guarantees the existence of a perfect matching in $G_b$ for the first $k$ nodes $\{v_i | i \leq k\} \subseteq \mathcal{V}_{exit}$. This can be simply fulfilled by running the $\mathcal{O}(m^3)$ ($m = \max\{|\mathcal{V}_{exit}|, |\mathcal{V}_{pursuer}|\}$) Hungarian Algorithm for each $k \leq |\mathcal{V}_{exit}|$. Intuitively, the existence of a perfect matching means that the closest $k$ exits to the evader can be blocked simultaneously.

- For the pursuers involved in the perfect matching, their current policies are to follow the shortest path towards the matched exits. For any remaining pursuer indicated by $v$, the current policy is to follow the shortest path towards the exit with the minimum index $i$ that satisfies $(v_i, v) \in E_b$. For the evader, if at least one exit is removed from $G_b$, its current policy is to follow the shortest path to the closest removed exit. Otherwise, its current policy is to follow the shortest path to the closest exit that has not been occupied by any pursuer.

Note that the heuristic algorithm encourages the pursuers to increase the (lower-bound) steps for a worst-case evader to reach an exit. This can be viewed as approximating the equilibrium policy via a minimax mechanism. Besides, since the heuristic evader can flexibly switch to a better exit when the game situation changes, it is also hard to exploit in practice.

Compared to the DP approach in no-exit scenarios, the heuristic approach can compute the current policy for each state independently. At the price of being more exploitable in certain handcrafted cases, the heuristic policy can be directly computed during RL training, with a time complexity polynomial in the agent number for any current state. Since it avoids a preprocessing stage that traverses the state space, the heuristic approach also guarantees a better applicability of our EPG framework in multi-exit scenarios.

# C  Training Details

## C.1  Implementation Details

For the training of Q functions, which is not specified in Section 3.3, we employ the common technique of double target networks to avoid overestimations [13]. When there are two or three pursuers, we simply use centralized value networks to directly represent Q functions. In the scenarios with more pursuers, we use value-decomposition networks [28] as a simple way to decompose the joint Q functions. As the pursuers are homogeneous agents, we have a direct decomposition $Q_\phi(s, a) = \sum_{i=1}^{m} Q_\phi(s, a_i)$, where $m$ is the number of the pursuers.

In either the no-exit or multi-exit scenario, we use the same hyperparameter setting shown in Table 3 throughout the EPG training for either pursuer or evader policy. Note that the target entropy $\overline{H}$ in SAC [6] is defined in the form of $-\log(1/|\mathcal{A}|) = \log|\mathcal{A}|$ multiplied by a predetermined coefficient.

Table 3: Hyperparameter setting

| discount factor $\gamma$ | 0.99 |
|---|---|
| embedding dimension $d$ | 128 |
| number of attention heads | 8 |
| equilibrium guidance coefficient $\beta$ | 0.1 |
| pursuer target entropy coefficient | 0.05 |
| evader target entropy coefficient | 0.1 |
| batch size | 128 |
| learning rate | $10^{-5}$ |
| update epoch | 8 |

## C.2  Learning Curves

Besides the 2-pursuer case shown in the main paper, we have also verified the EPG framework in the 3-pursuer no-exit scenario using the exact equilibrium oracle from the DP approach (Algorithm 1). The learning curves of pursuer policy are shown in Figure 5.

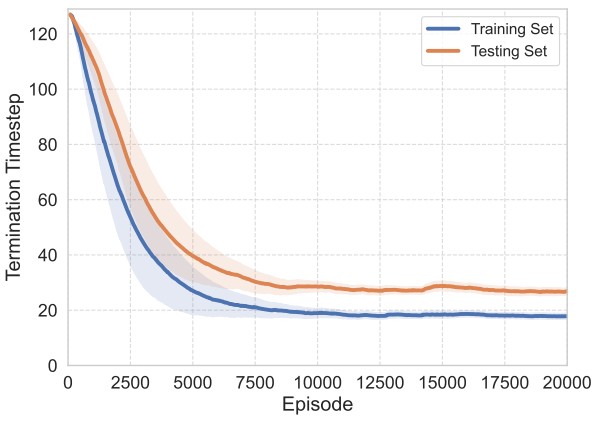

Figure 5: Pursuer learning curves in 3-pursuer no-exit PEGs

Figure 6 provides the learning curves of pursuer policy when trained with the grouping-extended DP approach in the 6-pursuer scenario. Note that the difficulty of pursuit is not reduced when it comes to 6 pursuers since a success pursuit requires 3 pursuers simultaneously adjacent to the evader. Also note that our RL policy eventually demonstrates a better performance in the testing set than in the training set. This shows the strong zero-shot generalizability of our approach in the 6-pursuer scenario and also explains the strong performance of the RL pursuer in Table 5.

Figure 7 shows the training process of the evaders in 2-pursuer and 6-pursuer scenarios, respectively. In contrast to pursuers, the evader aims to maximize the termination timesteps in no-exit scenarios.

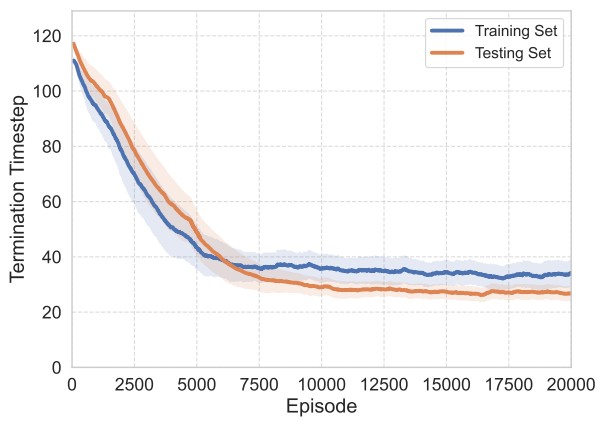

Figure 6: Pursuer learning curves in 6-pursuer no-exit PEGs

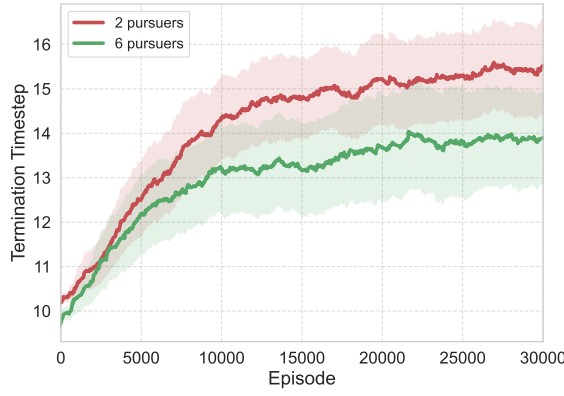

Figure 7: Evader learning curves in 2-pursuer and 6-pursuer scenarios

Note that in our practical implementation, the pursuer side receives a positive reward of $+30$ when capturing the evader, which aligns with the hyperparameters in Table 3. During training of the evader, a reward of $-30$ is in turn received upon pursuit, which corresponds to the adversarial game setting.

### C.3  Compute Resources and Time Costs

For no-exit graphs, EPG requires preprocessing the Nash value (or, equivalently, the array $D$ in the DP algorithm) for each graph contained in the training set. Since our DP algorithm has a near-optimal time complexity, for a 2-pursuer 500-node graph, it only takes around 10 seconds to run Algorithm 1 without the last loop, using a single 12th Gen Intel Core i7-12700F CPU. For multi-exit graphs, the matching-based heuristic algorithm is directly executed online and has no preprocessing requirement.

For the reinforcement learning process, Table 4 provides our recorded time for running 1000 EPG episodes, using two NVIDIA A100-SXM4-40GB GPUs. Intuitively, the training time grows sublinearly in the number of agents involved in the policy.

Table 4: Time requirement for 1000 EPG episodes

| Game Scenario | Training Object | Time |
|---|---|---|
| 2-pursuer no-exit PEG | 2-pursuer policy | 99 min |
| 6-pursuer no-exit PEG | 6-pursuer policy | 180 min |
| 2-pursuer no-exit PEG | evader policy | 75 min |
| 6-pursuer no-exit PEG | evader policy | 75 min |
| 5-pursuer 8-exit PEG [17] | 5-pursuer policy | 159 min |

# D Experimental Details

## D.1 Further Tests in Real-World Graphs

Figure 8 provides the illustrations of the real-world graphs used in Section 4.1.

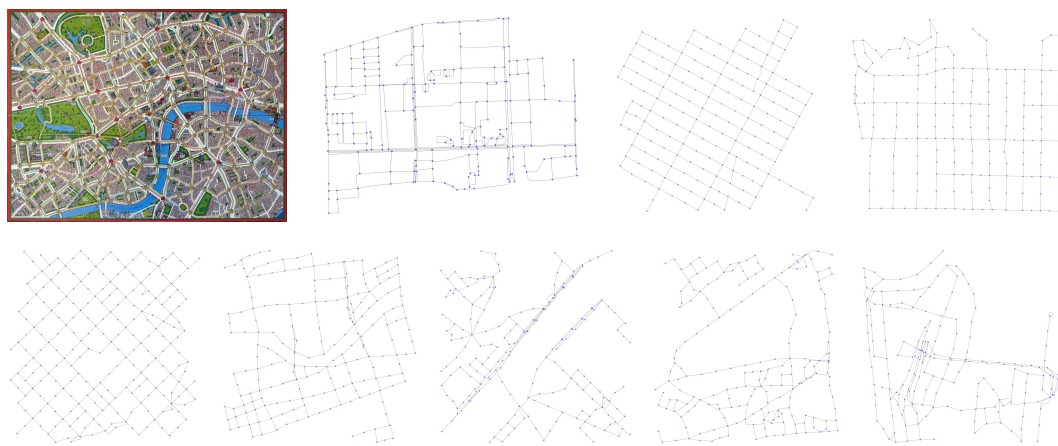

Figure 8: Illustrations of Scotland-Yard Map, Downtown Map, and 7 famous real-world locations (following the order in Tables 1 and 5)

Table 5: Performance of RL pursuer / evader against DP oracle in no-exit PEGs with 6 pursuers

| Graph Structure | Pursuit Success Rate ↑ | | | Evasion Timestep ↑ | |
|---|---|---|---|---|---|
| | DP - DP | RL - DP | SPS - DP | DP - DP | DP - RL |
| Grid Map | 1.00 | 1.00 | 1.00 | $7.73 \pm 2.75$ | $6.28 \pm 2.57$ |
| Scotland-Yard Map | 1.00 | 0.98 | 0.00 | $10.28 \pm 3.70$ | $7.60 \pm 2.77$ |
| Downtown Map | 1.00 | 0.98 | 0.01 | $10.44 \pm 3.37$ | $7.77 \pm 2.83$ |
| Times Square | 1.00 | 0.99 | 0.01 | $18.06 \pm 6.67$ | $14.54 \pm 4.84$ |
| Hollywood Walk of Fame | 1.00 | 0.95 | 0.00 | $34.90 \pm 16.32$ | $24.15 \pm 10.20$ |
| Sagrada Familia | 1.00 | 0.96 | 0.00 | $27.11 \pm 11.77$ | $19.11 \pm 5.90$ |
| The Bund | 1.00 | 0.91 | 0.02 | $32.14 \pm 16.98$ | $22.01 \pm 7.95$ |
| Eiffel Tower | 1.00 | 0.99 | 0.21 | $26.15 \pm 9.29$ | $20.96 \pm 7.70$ |
| Big Ben | 1.00 | 0.96 | 0.00 | $32.61 \pm 14.09$ | $24.45 \pm 9.77$ |
| Sydney Opera House | 1.00 | 0.85 | 0.00 | $33.23 \pm 13.56$ | $24.60 \pm 10.88$ |

Table 5 presents the results of our further tests in the 6-pursuer no-exit PEGs under the real-world graphs. As is shown in the table, the pursuit is still guaranteed by the grouped DP policy. This is reasonable because every sub-team of 2 pursuers guarantees the eventual adjacency for half of the pursuers (i.e., one pursuer). Besides, our MARL pursuer exhibits a high rate of success against the DP-based evader under the grouping mechanism.

The shortest path strategy (SPS), however, has very low success rates against the DP-based evader in the real-world graphs. This reflects the strength of our DP policies even under a grouping extension that trades off the optimality. Besides, our RL evader maintains a stable performance against the DP pursuer, with an acceptable decline of evasion steps as in the 2-pursuer case.

## D.2 Test Details

As is stated in Section 4, our zero-shot testing graphs are differentiated from the 152 Dungeon training graphs: Section 4.1 (Table 1) uses 10 unseen real-world graph structures. Section 4.2 (Figure 3) uses 10 unseen Dungeon graphs. Section 4.3 (Figure 4) uses the 10 unseen Dungeon graphs (left, zero-shot evaluations during training), Grid Map (mid), and Scotland-Yard Map (right).

For the tests in no-exit PEGs, the initial positions for the agents in the testing graphs are randomly generated under the restriction that the shortest path distance between the pursuers and the evader is greater than 5. This restriction serves to rule out certain easy pursuits.

For multi-exit PEGs, we randomly generated 1000 initial positions for the exits and agents, strictly following the descriptions and code from Grasper [17] for Grid Map and Scotland-Yard Map. For example, the minimum length of the evader's shortest path to any exit is set to 6 for the Grid Map and 5 for the Scotland-Yard Map. While [17] claims that they rule out the trivial cases that are either too difficult or too simple, the specific restrictions on the initial conditions are related to their method and hard to replicate. Therefore, we simply rule out the cases where it is clearly impossible for the pursuers or the evader to win. For the evader, we require that its distance to the closest exit be no greater than 10 (the time horizon of the game). For the pursuers, we require that, for any exit that the evader can reach in 10 steps, there is at least one pursuer who is closer than or as close as the evader with respect to the shortest path distance towards the exit.

All of the tests for the zero-shot performance of our RL policies are conducted after 30000 EPG episodes (each with at most 128 transitions) within the training set that contains a total of 152 graphs.

### D.3 Pursuit and Evasion Examples in No-Exit Graphs

Figures 9 and 10 use a PEG example in the 2-pursuer no-exit scenario to illustrate how our RL policy succeeds in capturing the DP evader while SPS is stuck in a loop (from step 9 to step 29) under the same initial condition. The graph is abstracted from a procedurally generated Dungeon map unseen during the training process. The circles represent the pursuers, and the square represents the evader.

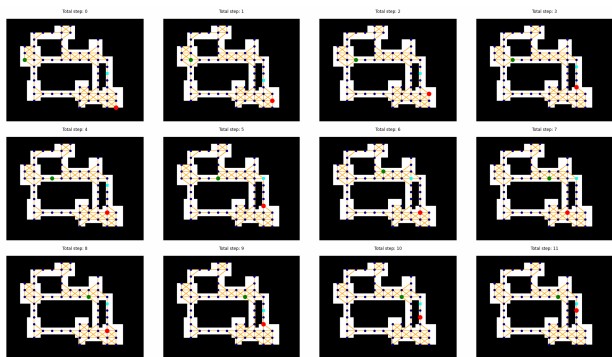

Figure 9: Successful pursuit against DP evader by RL pursuers

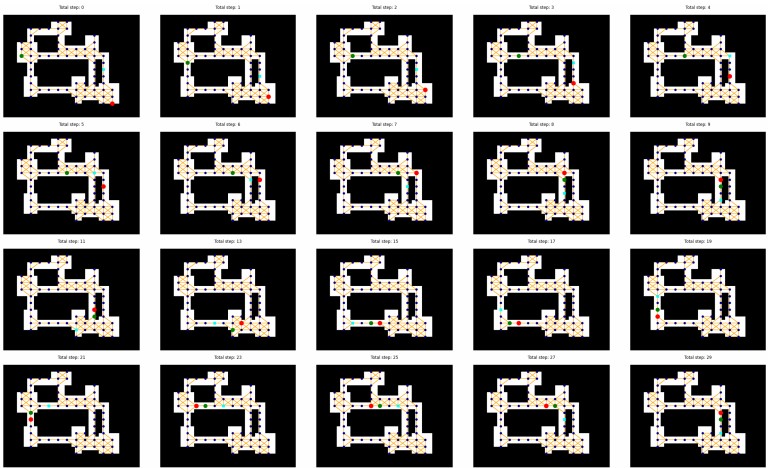

Figure 10: Failed pursuit against DP evader by shortest path strategy

Figure 11 shows a pursuit-evasion example between 6 grouped DP pursuers and the RL evader trained through EPG. The number of pursuers at the same node is explicitly shown on the circle when it is more than one. While surrounded by the pursuers, the RL evader seeks to maximize the timesteps before being captured by 3 adjacent pursuers (i.e., half of the pursuers).

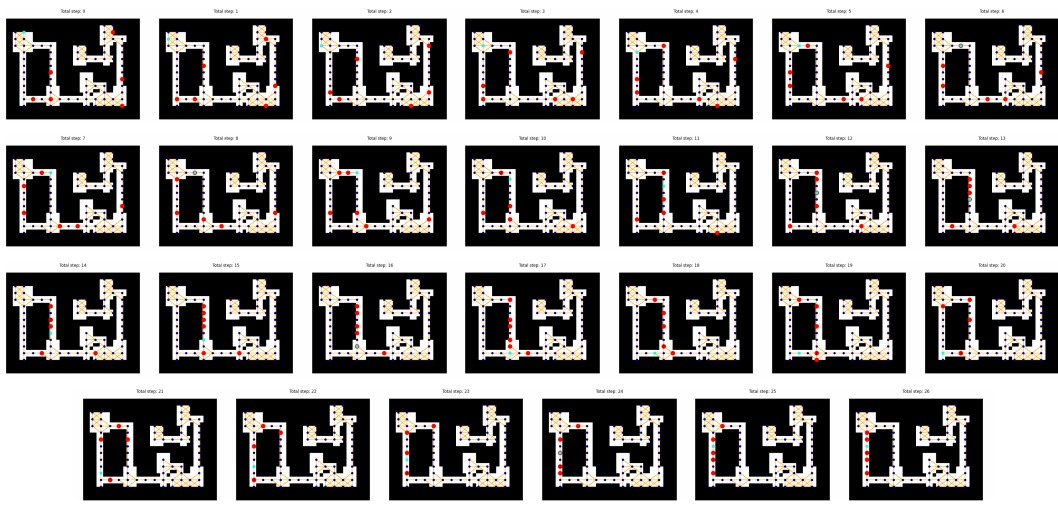

Figure 11: DP pursuers versus RL evader in 6-pursuer no-exit graph

Figure 12 shows how the RL evader manages to evade SPS pursuers just like the DP evader. Although the evader policy does not train against SPS, it manages to find how to create a loop that can make similar pursuer strategies fail. The emergence of such behaviors could also reflect the generalization capability of our EPG framework.

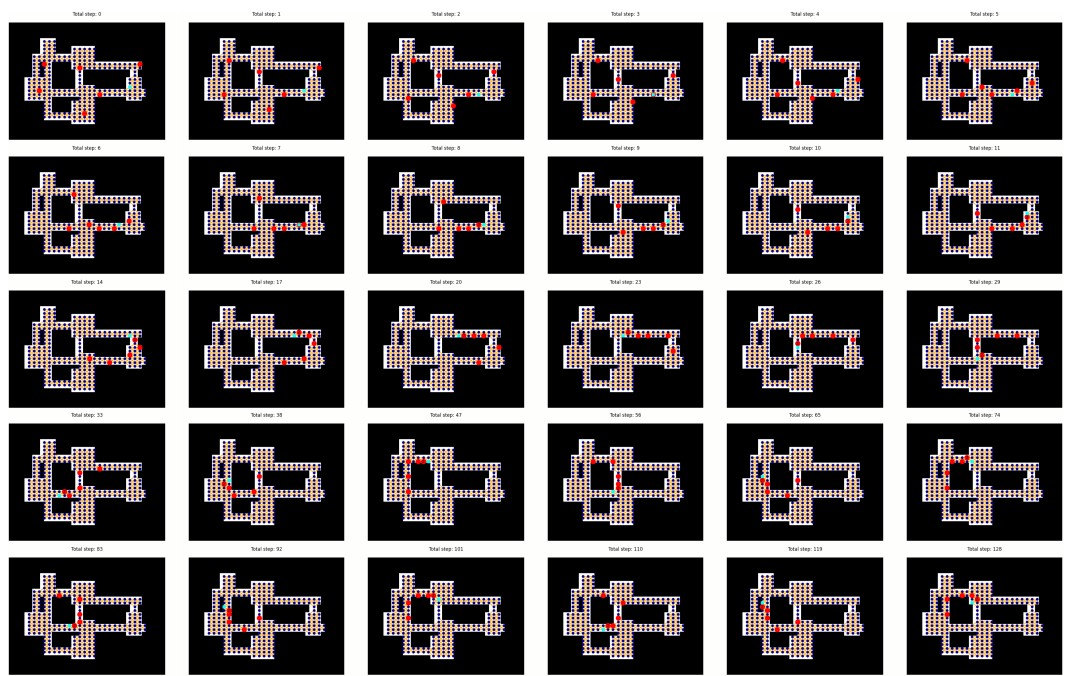

Figure 12: Shortest path strategy versus RL evader in 6-pursuer no-exit graph

### D.4 Pursuit Examples in Multi-Exit Graphs

While the time horizon for no-exit PEGs is 128, it is only set to 10 in multi-exit PEGs to align with [17]. Therefore, we show more pursuit examples in graphs with exits from the Dungeon environment. Figures 13, 14, and 15 provide three examples to show the zero-shot performance of our 5-pursuer policy trained through EPG with equilibrium heuristic. The light stars represent the 8 exits.

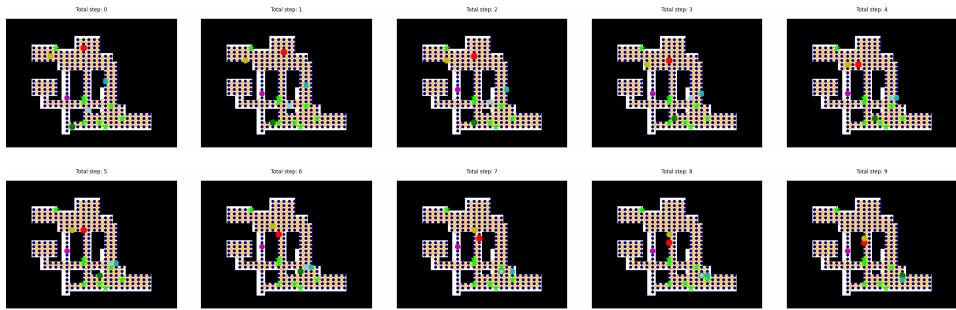

Figure 13: Pursuit example one in 5-pursuer 8-exit graph

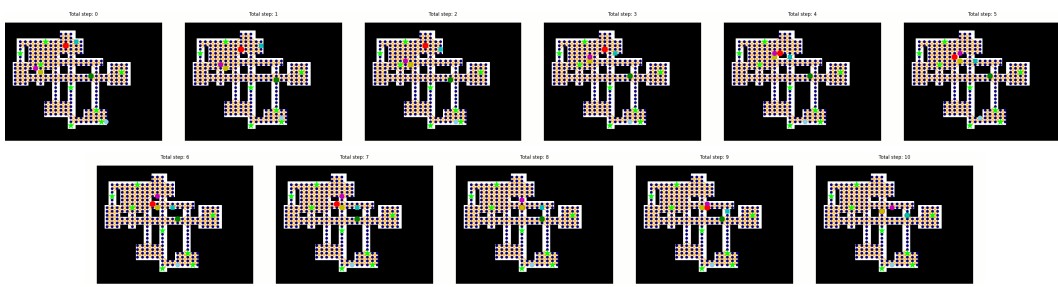

Figure 14: Pursuit example two in 5-pursuer 8-exit graph

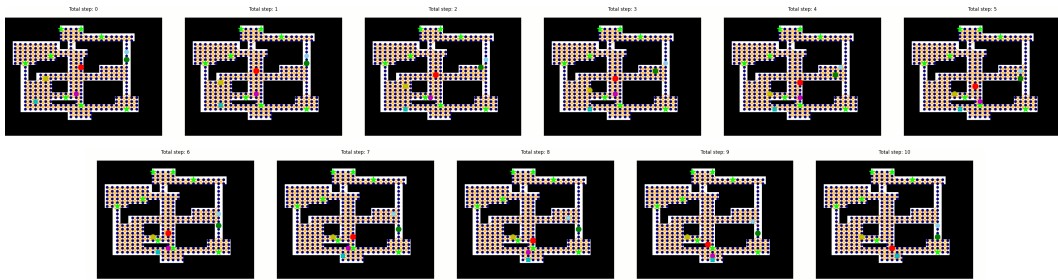

Figure 15: Pursuit example three in 5-pursuer 8-exit graph

Intuitively, in the first example, the two pursuers on the right manage to block the two exits in time. In the second example, the four pursuers near the evader manage to surround it from all directions. In the third example, the yellow pursuer guards one nearby exit, while the red pursuer and the purple pursuer cooperate to guard the other exit close to the evader.

# E Can Zero-Shot RL Pursuers Outperform DP?

While our previous results demonstrate the optimality of the DP policies, which maintain the pursuit success rate of 1 whenever possible, the mixed-strategy RL pursuers trained through EPG can actually outperform the pure-strategy DP pursuers when the pursuit is not globally guaranteed. We use the no-exit PEG results under two other real-world graph structures to show this.

## E.1 Results in Scale-Free Graph

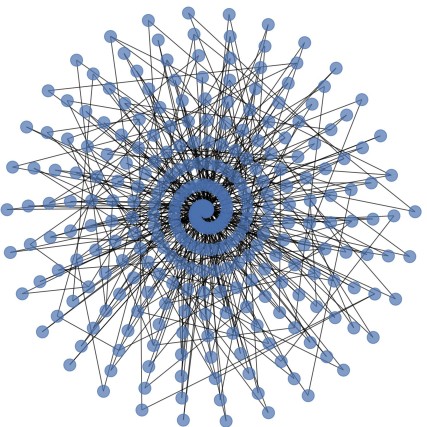

Figure 16: Illustration of Scale-Free Graph

Figure 16 provides an illustration of Scale-Free Graph, which is generated using Barabási-Albert preferential attachment [1]. With a maximum degree of 21 and a diameter of 11, Scale-Free Graph has very strong connectivity that can make evasion easier. As the pursuit is not guaranteed for 2 pursuers in this graph, we find that the DP pursuer policy only has a success rate of 0.1 against the DP evader. With a success rate of 0.44, our RL policy clearly outperforms DP under this graph.

## E.2 Results under Singapore Graph and Success Condition Shift

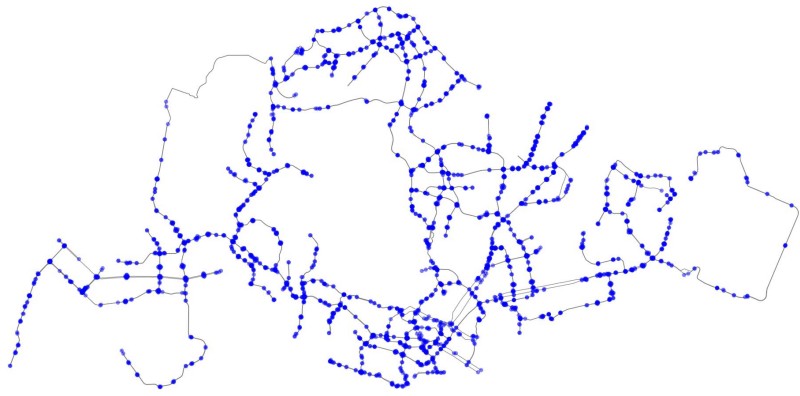

Figure 17: Illustration of Singapore Graph

Figure 17 provides an illustration of Singapore Graph, which is a high-level abstraction for the map of Singapore. With a maximum degree of 13 and a diameter of 41, the connectivity of the Singapore Graph is not as good as that of the Scale-Free Graph. While the pursuit is guaranteed for 2 pursuers under the success condition that one pursuer is adjacent to the evader, it is no longer guaranteed when we strengthen the condition to be both pursuers. In this case, the success rate of DP drops from 1 to 0.04, while the success rate of RL drops from 0.78 to 0.49. This comparative result suggests that the RL policy trained through our EPG framework could be robust against certain shifts of game rules.

Actually, we have further verified such robustness of our RL policy in the testing graphs that we used in the main paper. Table 6 compares the percentages of successful pursuit maintained by DP and RL pursuers after strengthening the success condition under the 10 graph structures. Compared with the DP pursuer, the RL pursuer demonstrates a clear advantage of robustness, with the maintenance rates over 60% (under 1000 tests for each graph) in 9 out of the 10 graphs.

Table 6: Percentages of successful pursuit maintained after success condition shift

| Graph Structure | DP pursuer | RL pursuer |
|---|---|---|
| Grid Map | 28.9% | 90.6% |
| Scotland-Yard Map | 27.9% | 90.4% |
| Downtown Map | 17.0% | 84.6% |
| Times Square, New York | 20.7% | 68.4% |
| Hollywood Walk of Fame, LA | 4.7% | 93.6% |
| Sagrada Familia, Barcelona | 1.7% | 78.1% |
| The Bund, Shanghai | 2.0% | 65.9% |
| Eiffel Tower, Paris | 9.1% | 40.3% |
| Big Ben, London | 6.3% | 66.1% |
| Sydney Opera House, Sydney | 8.6% | 79.5% |

# F  Additional Evaluations

## F.1  Performance under Large-Scale Graphs

Here we conduct additional experiments to further show that our trained policy can directly generalize to large-scale graphs. We increase both map range and discretization accuracy to derive significantly larger graphs for the eight real-world locations in Table 1. Specifically, the larger graphs are generated by doubling the discretization accuracy and increasing the map range. For Sydney Opera House, we actually double the map range to create a graph with at least 1000 nodes.

Table 7: Performance comparison across different maps at original and large scales

| Location | Original Scale | | Large Scale | |
|---|---|---|---|---|
| | Node Number | Success Rate | Node Number | Success Rate |
| Downtown Map | 206 | 0.99 | 907 | 0.99 |
| Times Square | 171 | 0.98 | 768 | 0.89 |
| Hollywood Walk of Fame | 201 | 0.62 | 892 | 0.71 |
| Sagrada Familia | 231 | 0.66 | 899 | 0.57 |
| The Bund | 200 | 0.60 | 952 | 0.54 |
| Eiffel Tower | 202 | 0.97 | 616 | 0.82 |
| Big Ben | 192 | 0.91 | 675 | 0.78 |
| Sydney Opera House | 183 | 0.74 | 1074 | 0.83 |

The success rates of two RL pursuers against the DP evader under the original graphs and the large graphs are compared in Table 7. While the node number increases significantly, the pursuit success rates do not suffer from a significant decline under large-scale graphs. In scenarios like Hollywood Walk of Fame and Sydney Opera House, they are even higher because of the structural changes.

Note that we do not fine-tune our policy on the larger graphs. We only adjust the termination function to keep the difficulty of the real-world pursuit task under the new graphs close to the original one. Specifically, one pursuer succeeds when its distance to the evader is no greater than 1 on the original graph and when the distance is no greater than 2 on the larger graph. Since the discretization accuracy doubles, the two conditions correspond to roughly the same real-world pursuit condition despite some local structural changes. Under the new graphs with the new success condition, we rerun the DP algorithm (Algorithm 1) to generate the optimized policies correspondingly.

### F.2 Performance against PSRO Policies

Besides DP, we also test the results of our EPG pursuer and evader policies against the PSRO [16] policies directly trained on the testing graphs. As is shown in Table 8, our EPG policies guarantee to capture the PSRO evader and survive significantly longer against the PSRO pursuers (except for Grid Map). This further verifies the benefit of using EPG for cross-graph zero-shot generalization.

Table 8: Performance comparison between EPG and PSRO policies

| Graph Structure | Termination Timestep [Success Rate] | |
| --- | --- | --- |
| | $EPG_p$ - $PSRO_e$ | $PSRO_p$ - $EPG_e$ |
| Grid Map | $14.04 \pm 6.88$ [1.00] | $11.87 \pm 2.39$ [1.00] |
| Scotland-Yard Map | $17.45 \pm 9.17$ [1.00] | $25.48 \pm 16.80$ [1.00] |
| Downtown Map | $14.14 \pm 7.37$ [1.00] | $38.39 \pm 30.50$ [0.97] |
| Times Square, New York | $17.72 \pm 7.86$ [1.00] | $37.87 \pm 29.94$ [0.97] |
| Hollywood Walk of Fame, LA | $30.93 \pm 17.67$ [1.00] | $60.71 \pm 39.21$ [0.86] |
| Sagrada Familia, Barcelona | $26.06 \pm 12.43$ [1.00] | $39.50 \pm 28.25$ [0.97] |
| The Bund, Shanghai | $24.76 \pm 13.01$ [1.00] | $34.97 \pm 24.75$ [0.99] |
| Eiffel Tower, Paris | $19.10 \pm 10.69$ [1.00] | $25.24 \pm 14.07$ [1.00] |
| Big Ben, London | $23.70 \pm 13.00$ [1.00] | $53.67 \pm 36.41$ [0.90] |
| Sydney Opera House, Sydney | $26.14 \pm 12.81$ [1.00] | $67.23 \pm 43.37$ [0.77] |

## G   Broader Impacts

As this work presents a general framework that leads to robust RL pursuer or evader policies with desirable zero-shot performance across different real-world graphs, it can be directly applied to real-world PEG solving under varying graph structures. The EPG framework in principle allows for an arbitrary amount of pursuers, exits, and graph nodes and thus guarantees applicability to diverse PEG settings in the real world. Since the inference time is negligible under GPUs, the trained policy model guarantees real-time applicability even when the graph structure dynamically changes, as we only need to recompute the shortest path distances as inputs.

From a security perspective, through our approach, a pursuit policy under urban streets can be readily generated for any traffic conditions that may lead to quite different graph connectivity. When trained under a more diverse set of graphs and a larger scale of parameters, a "large" policy model for general security purposes can be obtained. Such a model can save a huge amount of computation for solving different real-world PEGs. For example, while it can take hours to compute equilibrium policies in 500-node no-exit graphs with 3 pursuers through Algorithm 1, this time consumption is only required during preprocessing and training through EPG rather than during execution.

Also, our trained models can efficiently generate reasonable pursuit and evasion examples that may relate to real-world pursuit and evasion behaviors. This could help people better understand and predict the motivations of both the pursuers and the evader. Besides, due to its generality, the EPG training pipeline can also be extended for robust policy learning in other adversarial game domains. The idea of constructing equilibrium adversaries through preprocessed or preconstructed (accurate or approximate) oracles could have further impacts on subsequent reinforcement learning research.

