# OpenReview forum: "Equilibrium Policy Generalization: A Reinforcement Learning Framework for Cross-Graph Zero-Shot Generalization in Pursuit-Evasion Games"
_NeurIPS.cc/2025/Conference — NeurIPS 2025 poster_

### Official Review · Reviewer_LAVP · 2025-06-30

**Clarity:** 3
**Significance:** 4
**Originality:** 3
**Rating:** 4
**Confidence:** 3

**Summary:**

This paper proposes using reinforcement learning (RL) to learn pursuit-evasion game (PEG) policies that generalize across different graph environments. The authors simulate the adversary's policy using a dynamic programming (DP) algorithm for no-exit Markov PEGs and a heuristic for multi-exit cases. They then train a soft-actor-critic agent against this equilibrium adversary. Experiments on real-world graphs demonstrate zero-shot generalization of the learned policy.

**Questions:**

1) How does the RL algorithm perform when tested against non-equilibrium policies (e.g., random or heuristic agents)? Also, what is the performance of $\text{RL}_p$–$\text{RL}_e$ in Table 1?
2) Please provide more details on the graphs used in Table 1 (e.g., number, average degree, diameter). Also, the authors could explain why $\text{RL}_p$–DP underperforms on graphs like Hollywood Walk of Fame, Sagrada Familia, and The Bund, to help the readers have a full understand of the proposed method.
3) There is an inconsistency in Table 3: Line 572 mentions 1,000 trials, while the caption states 100 episodes. Please clarify the correct number. Additionally, since computational efficiency is a key contribution, it would be more informative to report and compare the average computation time per episode for EPG and DP in the main text rather than in the appendix.

**Ethical Concerns:**

["NO or VERY MINOR ethics concerns only"]

**Final Justification:**

Thanks to the authors for their response. The rebuttal has addressed some of my concerns regarding training details and the advantages of RL over DP. However, I still have reservations about applying the proposed EPG policies to more complex scenarios with larger agent scales. As noted in the rebuttal, grouping pursuers under six is more reasonable, yet in the results provided for Question 1, RL pursuers generally underperform compared to DP in the 2-pursuer case. However, I still acknowledge the strength of the proposed RL method in cross-graph generalization and therefore maintain my initial positive score.

**Limitations:**

The proposed method’s scalability is constrained by the shortest-path-based state representation, which incurs O($n^3$) complexity.

**Paper Formatting Concerns:**

None.

**Quality:**

2

**Strengths And Weaknesses:**

**Strengths**:
1) The paper shows that training RL agents against an equilibrium adversary on a graph corpus enables generalization to unseen graphs.
2) Experiments on diverse real-world maps demonstrate the method's strong zero-shot performance.
3) A sequential decision-making framework is proposed to address the large joint action space, enabling decentralized execution.

**Weaknesses**:
1) The claim that the RL method is computationally more efficient than traditional DP is not supported. The DP baseline reaches equilibrium in just 10 seconds (as mentioned in Line 569), raising questions about the RL method’s efficiency given its additional training cost.
2) The second claim—that the RL approach avoids overfitting to specific behaviors—would benefit from further evidence. As both training and testing rely on Nash equilibrium policies, it remains unclear how the method performs against other types of adversaries, such as random, SPS, or alternative RL agents in testing environments.
3) The training and test graphs are insufficiently differentiated, making the claimed generalization unclear.
4) The method is tested on a limited number of agents; its scalability to larger settings (e.g., >10 pursuers) is untested.

---

> ### Author Rebuttal · Authors · 2025-07-31
>
> Dear reviewer,
>
> Thank you very much for your review. We are glad to provide more details to clarify the weak points and answer your questions.
>
> **Weakness 1**
>
> > The claim that the RL method is computationally more efficient than traditional DP is not supported. The DP baseline reaches equilibrium in just 10 seconds (as mentioned in Line 569), raising questions about the RL method’s efficiency given its additional training cost.
>
> Thank you for the question on the RL method's efficiency. Actually, the DP algorithm that we propose in this paper already significantly improves over the traditional methods, guaranteeing a near-optimal time complexity of $\tilde{O}(|S|)$ (rather than $O(|S|^2)$). That said, our RL method has a range of advantages: First, when facing a new graph with 1000 nodes, DP will suffer from a significant slowdown (about 10 times slower than the 500-node case). In comparison, RL trained under 500-node graphs can immediately provide good policies, whose quality is verified by the scalability tests in our response to Reviewer BGj6. Under 1000 nodes, the necessary running of the $O(|V|^3)$ Floyd algorithm for RL is still finished within 1 second (since it has no extra constant factor), which is much faster than the $\tilde{O}(|V|^3)$ DP even under 2 pursuers. Second, in multi-exit cases, exact and efficient equilibrium oracles like DP are hard to construct. While simply following the heuristic pursuer policy to finish a game guarantees an average worst-case utility around 0.5 in our training corpus, the RL policy achieves an average worst-case utility over 0.65 after training. This additional result could further demonstrate the necessity of RL when only equilibrium heuristics are available.
>
> **Weakness 2**
>
> > The second claim—that the RL approach avoids overfitting to specific behaviors—would benefit from further evidence. As both training and testing rely on Nash equilibrium policies, it remains unclear how the method performs against other types of adversaries, such as random, SPS, or alternative RL agents in testing environments.
>
> Currently, we have conducted additional experiments against alternative RL agents like PSRO [1] and approximate best responses to our policies. The result is reserved in our response to Reviewer m6zW (Weakness 2) and further demonstrates the strength of EPG policies.
>
> **Weakness 3**
>
> > The training and test graphs are insufficiently differentiated, making the claimed generalization unclear.
>
> Thanks for mentioning the training and testing differentiation. As is stated in Lines 264-266, our training graphs are always the 152 Dungeon graphs. Our zero-shot testing graphs are differentiated from the training graphs: Section 4.1 (Table 1) uses 10 unseen real-world graph structures. Section 4.2 (Figure 3) uses 10 unseen Dungeon graphs. Section 4.3 (Figure 4) uses the 10 unseen Dungeon graphs (left, zero-shot evaluations during training), Grid Map (mid), and Scotland-Yard Map (right). Currently, we have added further clarifications in our paper.
>
> **Weakness 4**
>
> > The method is tested on a limited number of agents; its scalability to larger settings (e.g., >10 pursuers) is untested.
>
> Thank you for the comment on the agent number. There is actually a theoretical reason behind our choice of conducting experiments under 6 pursuers. [2] proves that in any planar graph, 3 pursuers are efficient to capture the evader. Therefore, when the pursuer number is overly large (e.g., >10 pursuers), it is less meaningful to learn a completely joint policy for real-world pursuit. A more reasonable substitute is to group the pursuers and use the 6-pursuer policies (or less) separately. For the evader side, it is also intuitively difficult to learn an effective evasion policy against too many pursuers. Besides, when compared to Table 1 and Figure 5, Table 4 and Figure 6 suggest that EPG could have even stronger zero-shot generalization capability in games with more agents.
>
> **Question 1**
>
> > How does the RL algorithm perform when tested against non-equilibrium policies (e.g., random or heuristic agents)? Also, what is the performance of $\mathrm{RL_{p}-RL_{e}}$ in Table 1?
>
> Currently, we have further tested EPG policies against PSRO [1]. Our policies guarantee to capture the PSRO evader and survive significantly longer against the PSRO pursuers. Besides, we find that directly training best responses against our policies does not lead to the policies that are better at exploiting ours in comparison with DP. The detailed results are reserved in our response to Reviewer m6zW.
>
> Here we provide the results of $\mathrm{RL_{p}-RL_{e}}$ for the "Evasion Timestep" in Table 1 and Table 4:
>
> | Map/Location | \| | 2-Pursuer Case | | | \| | 6-Pursuer Case | | |
> |---|---|---|---|---|---|---|---|---|
> | | \| | DP[p] vs DP[e] | DP[p] vs RL[e] | RL[p] vs RL[e] | \| | DP[p] vs DP[e] | DP[p] vs RL[e] | RL[p] vs RL[e] |
> | Grid Map | \| | 12.29 ± 2.06 | 11.88 ± 2.39 | 14.34 ± 5.08 | \| | 7.73 ± 2.75 | 6.28 ± 2.57 | 8.67 ± 4.38 |
> | Scotland-Yard Map | \| | 15.13 ± 2.77 | 12.57 ± 2.96 | 17.82 ± 7.31 | \| | 10.28 ± 3.70 | 7.60 ± 2.77 | 11.17 ± 6.04 |
> | Downtown Map | \| | 14.22 ± 3.27 | 11.83 ± 3.12 | 17.65 ± 8.72 | \| | 10.44 ± 3.37 | 7.77 ± 2.83 | 10.98 ± 5.44 |
> | Times Square, New York | \| | 16.47 ± 3.23 | 14.68 ± 3.11 | 20.29 ± 9.65 | \| | 18.06 ± 6.67 | 14.54 ± 4.84 | 14.79 ± 7.58 |
> | Hollywood Walk of Fame, LA | \| | 25.56 ± 5.03 | 20.00 ± 4.99 | 34.34 ± 20.18 | \| | 34.90 ± 16.32 | 24.15 ± 10.20 | 21.03 ± 10.99 |
> | Sagrada Familia, Barcelona | \| | 21.88 ± 4.82 | 17.89 ± 4.59 | 26.87 ± 11.00 | \| | 27.11 ± 11.77 | 19.11 ± 5.90 | 18.18 ± 8.30 |
> | The Bund, Shanghai | \| | 25.26 ± 6.17 | 20.59 ± 5.59 | 30.66 ± 17.27 | \| | 32.14 ± 16.98 | 22.01 ± 7.95 | 19.47 ± 9.46 |
> | Eiffel Tower, Paris | \| | 23.42 ± 6.48 | 18.47 ± 6.12 | 28.41 ± 16.80 | \| | 26.15 ± 9.29 | 20.96 ± 7.70 | 19.86 ± 10.04 |
> | Big Ben, London | \| | 27.89 ± 6.35 | 21.58 ± 6.38 | 33.24 ± 18.71 | \| | 32.61 ± 14.09 | 24.45 ± 9.77 | 21.31 ± 10.25 |
> | Sydney Opera House, Sydney | \| | 26.92 ± 5.89 | 22.37 ± 6.16 | 32.94 ± 14.72 | \| | 33.23 ± 13.56 | 24.60 ± 10.88 | 21.32 ± 10.78 |
>
> While the RL pursuers are generally weaker than DP in the 2-pursuer case, they actually outperform DP against the RL evader in part of the 6-pursuer scenarios, where DP only constructs an approximate oracle under grouping extension.
>
> **Question 2**
>
> > Please provide more details on the graphs used in Table 1 (e.g., number, average degree, diameter). Also, the authors could explain why $\mathrm{RL_{p}-DP}$ underperforms on graphs like Hollywood Walk of Fame, Sagrada Familia, and The Bund, to help the readers have a full understand of the proposed method.
>
> Here we provide more detailed information on the graphs used in Table 1:
>
> | Map/Location | \| | Node Number | \| | Average Degree | \| | Diameter | \| |
> |---|---|---|---|---|---|---|---|
> | Grid Map | \| | 100 | \| | 3.60 | \| | 18 | \| |
> | Scotland-Yard Map | \| | 200 | \| | 3.91 | \| | 19 | \| |
> | Downtown Map | \| | 206 | \| | 2.98 | \| | 19 | \| |
> | Times Square, New York | \| | 171 | \| | 2.58 | \| | 22 | \| |
> | Hollywood Walk of Fame, LA | \| | 201 | \| | 2.42 | \| | 31 | \| |
> | Sagrada Familia, Barcelona | \| | 231 | \| | 2.60 | \| | 25 | \| |
> | The Bund, Shanghai | \| | 200 | \| | 2.53 | \| | 29 | \| |
> | Eiffel Tower, Paris | \| | 202 | \| | 2.34 | \| | 38 | \| |
> | Big Ben, London | \| | 192 | \| | 2.48 | \| | 34 | \| |
> | Sydney Opera House, Sydney | \| | 183 | \| | 2.33 | \| | 37 | \| |
>
> In planar graphs, a larger average degree generally implies the existence of small cycles. For example, in Grid Map, all minimal cycles' length is 4. Since successful evasions rely more on large cycles, graphs like Grid Map, Scotland-Yard Map, and Downtown Map are easier for pursuit. Note that while the Scale-Free Graph in Appendix E.1 has a very large average degree, the cycles in the graph are large as well since it is not a planar graph. Besides, Eiffel Tower, Big Ben, and Sydney Opera House all have large diameters, which implies the existence of long "branches" that have poor connectivity with other nodes (see Figure 8). Therefore, these graphs also benefit pursuit since the structures are closer to trees.
>
> Since Hollywood Walk of Fame, Sagrada Familia, and The Bund do not have the mentioned characteristics, these graphs are relatively hard for pursuers. This accounts for the comparatively low success rates of RL and SPS pursuers on these graphs. While Times Square shares a similar structure with these three graphs (also see Figure 8), the success rate of the RL pursuer is high. This suggests that our RL agents could have discovered near-optimal strategies even for some graph structures that do not ease pursuit.
>
> **Question 3**
>
> > There is an inconsistency in Table 3: Line 572 mentions 1,000 trials, while the caption states 100 episodes. Please clarify the correct number. Additionally, since computational efficiency is a key contribution, it would be more informative to report and compare the average computation time per episode for EPG and DP in the main text rather than in the appendix.
>
> Thanks for this comment. We are sorry for the typo on the caption and have made a revision. The correct number is 1,000 rather than 100. In the main paper, we have also added a comparison between the execution time of DP and RL and reported the average training time cost for EPG.
>
> Thank you again for providing the detailed review and helpful suggestions. We hope our response properly addresses your concerns. We are looking forward to having further discussions with you.
>
> **References**
>
> [1] Marc Lanctot, Vinicius Zambaldi, Audrunas Gruslys, Angeliki Lazaridou, Karl Tuyls, Julien Pérolat, David Silver, and Thore Graepel. A unified game-theoretic approach to multiagent reinforcement learning. Advances in Neural Information Processing Systems, 30, 2017.
>
> [2] Fromme and M. Aigner. A game of cops and robbers. Discrete Appl. Math, 8:1–12, 1984.

---

> > ### Comment · Reviewer_LAVP · 2025-08-06
> >
> > Thanks to the authors for their response. The rebuttal has addressed some of my concerns regarding training details and the advantages of RL over DP. However, I still have reservations about applying the proposed EPG policies to more complex scenarios with larger agent scales. As noted in the rebuttal, grouping pursuers under six is more reasonable, yet in the results provided for Question 1, RL pursuers generally underperform compared to DP in the 2-pursuer case. However, I still acknowledge the strength of the proposed RL method in cross-graph generalization and therefore maintain my initial positive score.

---

> > > ### Author Response · Authors · 2025-08-06
> > >
> > > Thank you again for reviewing our paper and providing the valuable comments!

---

### Official Review · Reviewer_m6zW · 2025-07-02

**Clarity:** 4
**Significance:** 3
**Originality:** 3
**Rating:** 5
**Confidence:** 4

**Summary:**

This work describes a framework to generally solve pursuit-evasion graph games. These are games where pursuer and evader agents exist on a graph and the pursuers must catch the evaders in a zero-sum setting. The authors propose an equilibrium policy generation method to learn generalized policies than can work zero-shot on unseen graph structures. They train RL agents against oracle policies which are discovered using dynamic programming.

My current rating is borderline reject. I would be willing to increase this score is better statistical data is provided, and if the paper is updated with a proper comparison of EPG in the no-exit scenarios against appropriate baselines, or if at least a good justification is provided for not comparing against them.

**Questions:**

1. Why are there no deviation data provided for the data in Table 1? additionally, what do the error bars in Figure 3 represent? There is a huge difference in the significance of standard error, standard deviation, and confidence intervals.
2. Why are the testing generalization plots in Figure 4 plotted over time in seconds? Why not flops, or timesteps? This would make for a computationally fair comparison.

**Ethical Concerns:**

["NO or VERY MINOR ethics concerns only"]

**Final Justification:**

Additional experiments, improved statistical data, references, and a better proof in the main body of the work have convinced me that the work is in an acceptable state to be recommended for acceptance.

**Limitations:**

See above

**Quality:**

3

**Strengths And Weaknesses:**

Strengths
1. The use of RL in this problem is well-motivated, as existing methods for PEGs are computationally intensive to fine-tune, and are not guaranteed to generalize across different graph structures.
2. The construction of oracle policies for training using DP is well-explained: the authors walk through a logical progression of the near-optimality of their DP approach with no-exit graphs, then an extension to exits, and then an extension to many pursuers. I have not rigorously checked the proof of Theorem 1.
3. The use of an attention mechanism to encode the observations of changing graph structures is well explained.
4. The experiments evince good results for EPG with the proposed training scheme and policy loss and distance features.

Weaknesses:
1.  Definition of the NE on pg 3 should include 'at all states'. Also, a Markov game must be finite to guarantee the existence of a NE. Suitable citations should also be provided for the Nash value and the 2 last claims in the NE paragraph.
2. The experiments are unfortunately a bit lacking as the authors only present ablations of their own method and comparisons to the DP solution. While the use of RL was well-motivated as other PEG solving algorithms would be difficult to generalize, it still would be useful to see if a naive implementation of PSRO in no-exit PEGs.
3. As the authors mention in the conclusion, a test with partial observability would also greatly improve the viability of this method, since this is a typical assumption in MARL. Since the observation encodings use an attention mechanism, it should not be too difficult to implement this.

---

> ### Author Rebuttal · Authors · 2025-07-31
>
> Dear reviewer,
>
> Thank you very much for reviewing our paper and providing the detailed comments. Here is our response to the weak points as well as your questions.
>
> **Weakness 1**
>
> > Definition of the NE on pg 3 should include 'at all states'. Also, a Markov game must be finite to guarantee the existence of a NE. Suitable citations should also be provided for the Nash value and the 2 last claims in the NE paragraph.
>
> Thank you for this comment. We have now included 'at all states' in the NE definition and specified the restrictions of finite states and actions. We have also added reference [1] for the Nash value and reference [2] for the last 2 claims in the NE paragraph.
>
> **Weakness 2**
>
> > The experiments are unfortunately a bit lacking as the authors only present ablations of their own method and comparisons to the DP solution. While the use of RL was well-motivated as other PEG solving algorithms would be difficult to generalize, it still would be useful to see if a naive implementation of PSRO in no-exit PEGs.
>
> Thanks for this suggestion. Currently, we have additionally tested the result of our RL pursuer and evader policies against PSRO policies directly trained on the testing graphs (10 iterations, 1000 episodes per iteration):
>
> | Map/Location | \| | Timestep (Success Rate) | | | |
> |---|---|---|---|---|---|
> | | \| | RL[p] vs PSRO[e] | \| | PSRO[p] vs RL[e] | |
> | Grid Map | \| | 14.04 ± 6.88 (1.00) | \| | 11.87 ± 2.39 (1.00) | |
> | Scotland-Yard Map | \| | 17.45 ± 9.17 (1.00) | \| | 25.48 ± 16.80 (1.00) | |
> | Downtown Map | \| | 14.14 ± 7.37 (1.00) | \| | 38.39 ± 30.50 (0.97) | |
> | Times Square, New York | \| | 17.72 ± 7.86 (1.00) | \| | 37.87 ± 29.94 (0.97) | |
> | Hollywood Walk of Fame, LA | \| | 30.93 ± 17.67 (1.00) | \| | 60.71 ± 39.21 (0.86) | |
> | Sagrada Familia, Barcelona | \| | 26.06 ± 12.43 (1.00) | \| | 39.50 ± 28.25 (0.97) | |
> | The Bund, Shanghai | \| | 24.76 ± 13.01 (1.00) | \| | 34.97 ± 24.75 (0.99) | |
> | Eiffel Tower, Paris | \| | 19.10 ± 10.69 (1.00) | \| | 25.24 ± 14.07 (1.00) | |
> | Big Ben, London | \| | 23.70 ± 13.00 (1.00) | \| | 53.67 ± 36.41 (0.90) | |
> | Sydney Opera House, Sydney | \| | 26.14 ± 12.81 (1.00) | \| | 67.23 ± 43.37 (0.77) | |
>
> As is shown in the table above, our RL policies guarantee to capture the PSRO evader and survive significantly longer against the PSRO pursuers (except for Grid Map). This further verifies the benefit of using EPG for cross-graph zero-shot generalization. Besides, we have also approximated the best responses (worst-case opponents) through reinforcement learning against EPG policies on the testing graphs. The approximate best-response policies are generally not better at exploiting EPG policies in comparison with DP:
>
> | Map/Location | \| | Pursuit Success Rate | | \| | Evasion Timestep | |
> |---|---|---|---|---|---|---|
> | | \| | RL[p] vs DP[e] | RL[p] vs BR[e] | \| | DP[p] vs RL[e] | BR[p] vs RL[e] |
> | Grid Map | \| | 1.00 | 1.00 | \| | 11.88 ± 2.39 | 11.67 ± 2.60 |
> | Scotland-Yard Map | \| | 0.99 | 1.00 | \| | 12.57 ± 2.96 | 13.12 ± 3.65 |
> | Downtown Map | \| | 0.99 | 0.99 | \| | 11.83 ± 3.12 | 13.81 ± 5.83 |
> | Times Square, New York | \| | 0.98 | 0.95 | \| | 14.68 ± 3.11 | 16.20 ± 5.16 |
> | Hollywood Walk of Fame, LA | \| | 0.62 | 0.67 | \| | 20.00 ± 4.99 | 20.22 ± 9.60 |
> | Sagrada Familia, Barcelona | \| | 0.66 | 0.76 | \| | 17.89 ± 4.59 | 18.83 ± 6.30 |
> | The Bund, Shanghai | \| | 0.60 | 0.56 | \| | 20.59 ± 5.59 | 21.91 ± 10.58 |
> | Eiffel Tower, Paris | \| | 0.97 | 0.98 | \| | 18.47 ± 6.12 | 19.28 ± 10.48 |
> | Big Ben, London | \| | 0.91 | 0.94 | \| | 21.58 ± 6.38 | 23.07 ± 12.91 |
> | Sydney Opera House, Sydney | \| | 0.74 | 0.80 | \| | 22.37 ± 6.16 | 25.78 ± 16.79 |
>
> **Weakness 3**
>
> > As the authors mention in the conclusion, a test with partial observability would also greatly improve the viability of this method, since this is a typical assumption in MARL. Since the observation encodings use an attention mechanism, it should not be too difficult to implement this.
>
> Thanks for the suggestion on the tests with partial observability. Actually, our subsequent studies have already verified that EPG also works in imperfect-information PEGs. The perfect-information DP oracle can be extended to imperfect-information scenarios by preserving beliefs on the opponent's position. Under either perfect-information or imperfect-information DP guidance, EPG under belief predictions can learn a generalized policy through the same cross-graph training paradigm as in perfect-information scenarios. While the actual learning curves converge at larger termination timesteps (over 30 under either guidance), the overall trends are similar to the black curves in Figure 3.
>
> **Question 1**
>
> > Why are there no deviation data provided for the data in Table 1? additionally, what do the error bars in Figure 3 represent? There is a huge difference in the significance of standard error, standard deviation, and confidence intervals.
>
> Thanks for the question. We have now rerun the experiments and updated the results with standard deviations for the "Evasion Timestep" columns in Table 1 (2 pursuers) and Table 4 (6 pursuers) under 500 tests. Following the suggestion from Reviewer LAVP, we have also added the result of $\mathrm{RL_{p}-RL_{e}}$ in the tables.
>
> | Map/Location | \| | 2-Pursuer Case | | | \| | 6-Pursuer Case | | |
> |---|---|---|---|---|---|---|---|---|
> | | \| | DP[p] vs DP[e] | DP[p] vs RL[e] | RL[p] vs RL[e] | \| | DP[p] vs DP[e] | DP[p] vs RL[e] | RL[p] vs RL[e] |
> | Grid Map | \| | 12.29 ± 2.06 | 11.88 ± 2.39 | 14.34 ± 5.08 | \| | 7.73 ± 2.75 | 6.28 ± 2.57 | 8.67 ± 4.38 |
> | Scotland-Yard Map | \| | 15.13 ± 2.77 | 12.57 ± 2.96 | 17.82 ± 7.31 | \| | 10.28 ± 3.70 | 7.60 ± 2.77 | 11.17 ± 6.04 |
> | Downtown Map | \| | 14.22 ± 3.27 | 11.83 ± 3.12 | 17.65 ± 8.72 | \| | 10.44 ± 3.37 | 7.77 ± 2.83 | 10.98 ± 5.44 |
> | Times Square, New York | \| | 16.47 ± 3.23 | 14.68 ± 3.11 | 20.29 ± 9.65 | \| | 18.06 ± 6.67 | 14.54 ± 4.84 | 14.79 ± 7.58 |
> | Hollywood Walk of Fame, LA | \| | 25.56 ± 5.03 | 20.00 ± 4.99 | 34.34 ± 20.18 | \| | 34.90 ± 16.32 | 24.15 ± 10.20 | 21.03 ± 10.99 |
> | Sagrada Familia, Barcelona | \| | 21.88 ± 4.82 | 17.89 ± 4.59 | 26.87 ± 11.00 | \| | 27.11 ± 11.77 | 19.11 ± 5.90 | 18.18 ± 8.30 |
> | The Bund, Shanghai | \| | 25.26 ± 6.17 | 20.59 ± 5.59 | 30.66 ± 17.27 | \| | 32.14 ± 16.98 | 22.01 ± 7.95 | 19.47 ± 9.46 |
> | Eiffel Tower, Paris | \| | 23.42 ± 6.48 | 18.47 ± 6.12 | 28.41 ± 16.80 | \| | 26.15 ± 9.29 | 20.96 ± 7.70 | 19.86 ± 10.04 |
> | Big Ben, London | \| | 27.89 ± 6.35 | 21.58 ± 6.38 | 33.24 ± 18.71 | \| | 32.61 ± 14.09 | 24.45 ± 9.77 | 21.31 ± 10.25 |
> | Sydney Opera House, Sydney | \| | 26.92 ± 5.89 | 22.37 ± 6.16 | 32.94 ± 14.72 | \| | 33.23 ± 13.56 | 24.60 ± 10.88 | 21.32 ± 10.78 |
>
> While the RL pursuers are generally weaker than the DP oracle in the 2-pursuer case, they actually outperform DP against the RL evader in part of the 6-pursuer scenarios, where DP only constructs an approximate oracle under grouping extension.
>
> For the error bars in Figure 3, we clarify that they represent standard deviations.
>
> **Question 2**
>
> > Why are the testing generalization plots in Figure 4 plotted over time in seconds? Why not flops, or timesteps? This would make for a computationally fair comparison.
>
> Thanks for the question. In Figure 4, the results besides EPG are cited from the paper of Grasper [3]. Since EPG requires no time for fine-tuning, we directly show the original time cost of the comparative methods reported in [3].
>
> Thank you again for the insightful comments and helpful suggestions. We hope our response properly addresses your concerns. We are looking forward to having further discussions with you.
>
> **References**
>
> [1] Lloyd S. Shapley. Stochastic games. Proceedings of the National Academy of Sciences, 39(10):1095–1100, 1953.
>
> [2] Tim Roughgarden. Twenty lectures on algorithmic game theory. Cambridge University Press, 2016.
>
> [3] Pengdeng Li, Shuxin Li, Xinrun Wang, Jakub Cerny, Youzhi Zhang, Stephen McAleer, Hau Chan, and Bo An. Grasper: A generalist pursuer for pursuit-evasion problems. In Proceedings of the 23rd International Conference on Autonomous Agents and Multiagent Systems, pages 1147–1155, 2024.

---

> > ### Comment · Reviewer_m6zW · 2025-08-06
> >
> > I thank the authors for their response to the reviews. After looking at the new results, I am more convinced by the statistical significance of the results and the veracity of the method. The experimental coverage is good and the updated definitions make the paper more readable. I am happy to raise my recommendation to accept as a result.

---

> > > ### Author Response · Authors · 2025-08-07
> > >
> > > Thank you again for providing the detailed review and helping us further improve this paper!

---

### Official Review · Reviewer_dxFV · 2025-07-03

**Clarity:** 4
**Significance:** 2
**Originality:** 3
**Rating:** 4
**Confidence:** 4

**Summary:**

This paper introduces the Equilibrium Policy Generalization (EPG) framework, a reinforcement learning method aimed at achieving robust zero-shot generalization for pursuit-evasion games (PEGs) across varying graph structures. The authors present a novel approach wherein reinforcement learning is guided by equilibrium policies obtained through dynamic programming (DP) or heuristic approximation. Specifically, an efficient DP algorithm is designed to find pure-strategy Nash equilibria for no-exit PEGs, and a heuristic based on bipartite matching is proposed for multi-exit scenarios. Experimental results demonstrate strong zero-shot performance and detailed ablation studies are provided to validate various components of the proposed method.

**Questions:**

- Have you tried using Graph Neural Networks (GNNs) to model the scenarios? If yes, how do they compare to your proposed sequence model with shortest path features? If no, can you clarify why such an approach was not considered?

- Why is the proposed approach limited specifically to pursuit-evasion games (PEGs)? It appears that the basic idea could also be applicable to more general two-player adversarial games (for example, two-player Texas Hold'em poker or simplified board games). Could you clarify the reasons behind focusing exclusively on PEGs?

- Could you further discuss the scalability of your equilibrium oracle methods (DP and heuristic)? Specifically, how realistic would it be to apply these methods to larger-scale games, and what practical considerations or limitations would arise?

**Ethical Concerns:**

["NO or VERY MINOR ethics concerns only"]

**Limitations:**

see weaknesses

**Quality:**

3

**Strengths And Weaknesses:**

Strengths

- The paper is well-organized, and the figures clearly illustrate the proposed ideas and framework.
- The core idea is intuitive and the motivation for equilibrium-guided reinforcement learning is well explained.
- The experiments and ablation studies are thorough and insightful, providing solid empirical evidence supporting the method’s effectiveness.

Weaknesses


- The proposed framework critically depends on the availability of an equilibrium opponent policy, which could be a strong assumption. Although the paper discusses two approaches (DP and heuristic) for constructing such equilibrium oracles, these solutions might be difficult to transfer to larger-scale or more general game scenarios, such as Atari environments or complex board/card games (e.g., chess).

- In essence, the method leverages a model-based or expert prior (guidance opponent), and thus the innovation from a purely RL-algorithm perspective is somewhat limited. Other RL methods such as PSRO, SAC, or even simple Q-learning could similarly benefit from having such a guidance opponent. Therefore, the comparisons presented in Figure 4 may not be entirely fair or balanced.

- The claimed zero-shot generalization capability is demonstrated primarily in scenarios with relatively modest variations. Even conventional RL methods, provided with a sufficiently large graph corpus (e.g., > 100 graphs), may achieve reasonable generalization. It would have been more convincing if the proposed approach could generalize beyond graph-based games.

---

> ### Author Rebuttal · Authors · 2025-07-31
>
> Dear reviewer,
>
> Thank you very much for reviewing our paper and providing the insightful comments. Here we provide a response to the weaknesses section and your questions.
>
> **Weakness 1**
>
> > The proposed framework critically depends on the availability of an equilibrium opponent policy, which could be a strong assumption. Although the paper discusses two approaches (DP and heuristic) for constructing such equilibrium oracles, these solutions might be difficult to transfer to larger-scale or more general game scenarios, such as Atari environments or complex board/card games (e.g., chess).
>
> Thank you for mentioning the transferability of our method. Actually, the EPG framework does not rely on an exact equilibrium oracle for good performance. In the multi-exit case, we only construct a heuristic that is intuitive and very fast to compute while still achieving current best results. The basic requirements of EPG are only twofold: First, the training corpus is diverse enough. Second, the opponents cannot be easily exploited. In principle, EPG is applicable to any other game scenarios through the use of general and approximate oracles like PSRO [1].
>
> **Weakness 2**
>
> > In essence, the method leverages a model-based or expert prior (guidance opponent), and thus the innovation from a purely RL-algorithm perspective is somewhat limited. Other RL methods such as PSRO, SAC, or even simple Q-learning could similarly benefit from having such a guidance opponent. Therefore, the comparisons presented in Figure 4 may not be entirely fair or balanced.
>
> Thank you for the comment on the innovation. Robust policy generalization in adversarial games requires considerations from a game-theoretic perspective. Simply optimizing against equilibrium opponents does not guarantee robust policies, which is also mentioned by Reviewer BGj6. As is clarified in our response to this point, EPG benefits from the compounding effect of SAC loss, cross-graph training, and equilibrium guidance. Essentially, EPG combines two complementary components, cross-graph/dynamics training and equilibrium adversary, each of which is not guaranteed to work individually. The robustness of EPG policies is assumed to benefit from training against the policies that are hard to exploit across a diverse set of graphs. This is quite different from learning against certain guidance policies in single dynamics. EPG essentially allows the generalized policy to train with a diverse set of opponents while avoiding overly exploiting certain behavior patterns. This eventually leads to a robust high-level policy across graphs.
>
> **Weakness 3**
>
> > The claimed zero-shot generalization capability is demonstrated primarily in scenarios with relatively modest variations. Even conventional RL methods, provided with a sufficiently large graph corpus (e.g., > 100 graphs), may achieve reasonable generalization. It would have been more convincing if the proposed approach could generalize beyond graph-based games.
>
> Thank you for the insight on zero-shot generalization. Here we should emphasize that policy generalization in multi-agent adversarial games like PEGs is more difficult than in single-agent cases. Recent work [2] mentions that even the state-of-the-art PEG methods (including Grasper [3], which already pretrains the policy in a set containing $1,000$ PEGs) struggle to adapt to rapid dynamic changes and do not guarantee cross-graph zero-shot performance. Our paper shows that robust cross-graph generalization requires proper adversaries (e.g., equilibrium policy) and high-level features (e.g., shortest path distance feature). Besides, the principle of sequential decision-making can give rise to a sequence model that maintains in-team cooperation during policy generalization. These findings may guide subsequent algorithm designs aimed at robust policy generalization in the broader domain of game RL.
>
> **Question 1**
>
> > Have you tried using Graph Neural Networks (GNNs) to model the scenarios? If yes, how do they compare to your proposed sequence model with shortest path features? If no, can you clarify why such an approach was not considered?
>
> Thanks for the question. Actually, the policy network part (on the right of Figure 2) of our sequence model is itself in the form of a graph neural network. The masked self-attention in the encoder has the same function as the aggregation layers in common GNNs. Furthermore, we use a novel shortest path distance feature as the initial node feature and embed the GNN into a multi-agent framework to form the sequence model. Without a GNN architecture, it would be difficult to encode the game states in graph-based PEGs.
>
> **Question 2**
>
> > Why is the proposed approach limited specifically to pursuit-evasion games (PEGs)? It appears that the basic idea could also be applicable to more general two-player adversarial games (for example, two-player Texas Hold'em poker or simplified board games). Could you clarify the reasons behind focusing exclusively on PEGs?
>
> Yes, the EPG framework can definitely be applied to other adversarial game settings. We focus on PEGs because they are quite important in real-world security but theoretically very hard to solve (EXPTIME-complete). Our proposed DP algorithm and grouping extension contribute to improving the scalability of existing methods with respect to both node number and agent number, while maintaining the theoretical foundation (Theorem 1). For more game types, general algorithms like PSRO can directly serve as equilibrium oracles in place of DP. However, applying them in PEGs does not guarantee the good performance of our approach since the game involves both cooperation among pursuers and competition between two teams.
>
> **Question 3**
>
> > Could you further discuss the scalability of your equilibrium oracle methods (DP and heuristic)? Specifically, how realistic would it be to apply these methods to larger-scale games, and what practical considerations or limitations would arise?
>
> Thanks for the question, and here we provide a further discussion. For the DP algorithm, our grouping extension guarantees its scalability with respect to agent number. When the node number is extremely large (e.g., 10000), we clearly need to function-approximate the $D(\cdot)$ table, which serves to construct the exact equilibrium oracle. For the heuristic algorithm, it is itself an approximate oracle and runs efficiently even in very large PEGs. Whether to use exact oracles like DP or approximate oracles like heuristic and PSRO in larger-scale games depends on the actual demand. If the game requires to be accurately solved, then the DP approach is a better choice. Besides, our additional experiments show that EPG trained under relatively small graphs can generalize well to larger-scale scenarios. This further enhances the applicability of DP in larger-scale games, as it is sufficient to use DP to presolve the games with moderate scales for training a policy that performs well in large scales.
>
> **Additional experiments**
>
> We increase both map range and discretization accuracy and derive significantly larger graphs for the real-world locations in Table 1. The pursuit success rate of $\mathrm{RL_{p}-DP}$ under the original graphs and the large graphs are shown as follows:
>
> | Map/Location | \| | Original Scale | | | \| | Large Scale | | |
> |---|---|---|---|---|---|---|---|---|
> | | \| | Node Number | \| | Success Rate | \| | Node Number | \| | Success Rate |
> | Downtown Map | \| | 206 | \| | 0.99 | \| | 907 | \| | 0.99 |
> | Times Square, New York | \| | 171 | \| | 0.98 | \| | 768 | \| | 0.89 |
> | Hollywood Walk of Fame, LA | \| | 201 | \| | 0.62 | \| | 892 | \| | 0.71 |
> | Sagrada Familia, Barcelona | \| | 231 | \| | 0.66 | \| | 899 | \| | 0.57 |
> | The Bund, Shanghai | \| | 200 | \| | 0.6 | \| | 952 | \| | 0.54 |
> | Eiffel Tower, Paris | \| | 202 | \| | 0.97 | \| | 616 | \| | 0.82 |
> | Big Ben, London | \| | 192 | \| | 0.91 | \| | 675 | \| | 0.78 |
> | Sydney Opera House, Sydney | \| | 183 | \| | 0.74 | \| | 1074 | \| | 0.83 |
>
> As is shown in the table, the success rates do not suffer from a significant decline under large-scale graphs. In scenarios like Hollywood Walk of Fame and Sydney Opera House, the success rates are even higher because of the structural changes.
>
> Thank you again for the insightful comments. We hope our response properly addresses your concerns. We are looking forward to having further discussions with you.
>
> **References**
>
> [1] Marc Lanctot, Vinicius Zambaldi, Audrunas Gruslys, Angeliki Lazaridou, Karl Tuyls, Julien Pérolat, David Silver, and Thore Graepel. A unified game-theoretic approach to multiagent reinforcement learning. Advances in Neural Information Processing Systems, 30, 2017.
>
> [2] Shuxin Zhuang, Shuxin Li, Tianji Yang, Muheng Li, Xianjie Shi, Bo An, and Youzhi Zhang. Solving urban network security games: Learning platform, benchmark, and challenge for AI research. arXiv preprint arXiv:2501.17559, 2025.
>
> [3] Pengdeng Li, Shuxin Li, Xinrun Wang, Jakub Cerny, Youzhi Zhang, Stephen McAleer, Hau Chan, and Bo An. Grasper: A generalist pursuer for pursuit-evasion problems. In Proceedings of the 23rd International Conference on Autonomous Agents and Multiagent Systems, pages 1147–1155, 2024.

---

### Official Review · Reviewer_BGj6 · 2025-07-03

**Clarity:** 3
**Significance:** 3
**Originality:** 2
**Rating:** 4
**Confidence:** 3

**Summary:**

This paper proposes the Equilibrium Policy Generalization (EPG) framework for pursuit-evasion games (PEGs), aiming to enable zero-shot generalization across graph structures. The core idea is to train reinforcement learning (RL) policies against equilibrium strategies of individual graphs using dynamic programming (DP) and a grouping mechanism for scalability. The authors claim that EPG outperforms state-of-the-art methods in zero-shot performance, supported by experiments on real-world graphs. While the framework shows promise, several critical limitations and theoretical gaps warrant discussion.

**Questions:**

See Weakness.

**Ethical Concerns:**

["NO or VERY MINOR ethics concerns only"]

**Final Justification:**

The authors have addressed all my concerns, thus I raise my score from 3 to 4.

**Limitations:**

Yes

**Paper Formatting Concerns:**

No.

**Quality:**

2

**Strengths And Weaknesses:**

**Strengths**
The work addresses a critical gap in real-world PEG applications: existing methods struggle with dynamic graph changes and require extensive fine-tuning, limiting real-time applicability. EPG’s focus on zero-shot generalization via equilibrium guidance is innovative and practical.

 **Weaknesses**
- The method requires pre-solving Nash equilibria for all training graphs, which is computationally intractable for large-scale scenarios. This limits applicability to real-world problems with dynamic or large graphs.

- The use of KL divergence between the current policy and precomputed equilibria as a loss raises a fundamental question: If the goal is to mimic equilibrium strategies, why not use direct supervised learning instead of RL? The paper lacks a clear justification for combining RL with equilibrium guidance over pure supervised training, which would be more efficient if equilibria are known.

- The training paradigm assumes that optimizing against Nash equilibria yields robust policies, but this is theoretically unsound. In zero-sum games, Nash equilibria are optimal against any strategy, but training an RL agent against an equilibrium does not guarantee the agent will converge to a NE. For instance, in rock-paper-scissors (a simple 3x3 game), RL training against this NE without constraints could yield arbitrary policies (e.g., a pure strategy).

 - The evaluation focuses on performance against precomputed NE, but the standard security metric for policies is *exploitability* (i.e., the maximum gain an opponent can achieve by best-responding to the policy). Failing to report exploitability results leaves uncertainty about whether EPG policies are truly robust or merely performant against a fixed NE.
 - The framework is evaluated in simplified scenarios , lacking scalability tests on large-scale graphs or complex real-world environments.

---

> ### Author Rebuttal · Authors · 2025-07-31
>
> Dear reviewer,
>
> Thank you very much for reviewing our paper. We definitely agree that it is important to clearly discuss the points in the weaknesses section. Here we provide our justifications and additional experiment results.
>
> **Point 1**
>
> > The method requires pre-solving Nash equilibria for all training graphs, which is computationally intractable for large-scale scenarios. This limits applicability to real-world problems with dynamic or large graphs.
>
> Thanks for the question about the real-world applicability of our method. In Line 158, we mention that directly solving PEGs requires time exponential in the agent number. Nevertheless, when the pursuer number is no more than 3, the time complexity of our DP algorithms is at most $\tilde{O}(|V|^4)$. Using CUDA acceleration, the computation for a 3-pursuer 500-node graph can be completed in several minutes. Also, EPG does not rely on the exact computation of Nash equilibrium. When the pursuer number is more than 3, our grouping extension decomposes the problem to the cases with 2 or 3 pursuers, guaranteeing the same time complexity and a desirable performance (Table 4). For multi-exit cases, we actually employ a very fast heuristic algorithm in place of exact equilibrium oracles while achieving the best results (Figure 4). Our additional experiments in response to the weakness on scalability (Point 5) further show that policies trained under relatively small graphs can generalize to large-scale scenarios through EPG. This could further reduce the need of pre-solving Nash equilibrium for the graphs with a large scale.
>
> **Point 2**
>
> > The use of KL divergence between the current policy and precomputed equilibria as a loss raises a fundamental question: If the goal is to mimic equilibrium strategies, why not use direct supervised learning instead of RL? The paper lacks a clear justification for combining RL with equilibrium guidance over pure supervised training, which would be more efficient if equilibria are known.
>
> Thanks for the question on supervised learning. In the experiments, we have tried mere supervised learning and found that it is consistently outperformed by RL with equilibrium guidance. In Section 4.2, we compare RL with mere supervised learning (blue lines in Figure 3), which suffers from a clear performance drop in no-exit scenarios. In Figure 4 (left), we show that mere supervised learning (green line) simply fails to generalize in multi-exit scenarios. Besides, even without equilibrium guidance, RL can still find good policies eventually (orange lines in Figure 3). These results suggest that RL is a better choice for cross-graph policy generalization than simply mimicking equilibrium strategies. Existing research in domains like LLMs [1] also suggests that RL has significantly better generalization capability than SL.
>
> **Point 3**
>
> > The training paradigm assumes that optimizing against Nash equilibria yields robust policies, but this is theoretically unsound. In zero-sum games, Nash equilibria are optimal against any strategy, but training an RL agent against an equilibrium does not guarantee the agent will converge to a NE. For instance, in rock-paper-scissors (a simple 3x3 game), RL training against this NE without constraints could yield arbitrary policies (e.g., a pure strategy).
>
> Thank you for providing this theoretical insight. We definitely agree that simply optimizing against equilibrium does not guarantee robust policies in a game. However, our approach has multiple differences: First, we establish EPG upon an entropy-regularized RL algorithm, SAC, which applies an indirect policy constraint that discourages pure strategies. For an RPS game, the policy regularization actually encourages the convergence to the NE $(1/3,1/3,1/3)$. Second, the robustness of EPG policies is assumed to benefit from training against the policies that are hard to exploit **across a diverse set of graphs**. This is quite different from learning against NE in single dynamics. While the preprocessed equilibrium adversaries can be generated by the same algorithm, they are essentially different policies under different dynamics. Therefore, EPG allows the generalized policy to train with a diverse set of opponents while avoiding overly exploiting certain behavior patterns. This eventually leads to a robust high-level policy across graphs. Third, the equilibrium guidance further facilitates efficient exploration and keeps the single-graph behavior of the generalized policy close to NE. From our understanding, these elements will compound to make the learned policy itself hard to exploit.
>
> **Point 4**
>
> > The evaluation focuses on performance against precomputed NE, but the standard security metric for policies is exploitability (i.e., the maximum gain an opponent can achieve by best-responding to the policy). Failing to report exploitability results leaves uncertainty about whether EPG policies are truly robust or merely performant against a fixed NE.
>
> Thank you for mentioning the exploitability of EPG policies. In our experiments for multi-exit scenarios in Figure 4 (mid & right), we report the utility against the worst-case opponent (higher value meaning lower exploitability) and compare with the state of the art under the same metric. The worst-case utility of the pursuer policies in the multi-exit tests can be exactly computed because the policy space of the evader is restricted to align with Grapser [2] (i.e., choosing an exit and following the shortest path to it). When there is no reasonable way to simplify the opponent (e.g., in no-exit scenarios), computing an exact best response can be intractable. In our experiments for no-exit scenarios, we use the DP policies as a substitute since they are near-optimal and, at the same time, effectively punish the policies that are not optimal. Actually, we have also approximated the best responses to EPG policies through reinforcement learning on the testing graphs. The approximate best-response (BR) policies are generally hard to train (since EPG policies are fairly strong) and not better at exploiting our policies in comparison with DP:
>
> | Map/Location | \| | Pursuit Success Rate | | \| | Evasion Timestep | |
> |---|---|---|---|---|---|---|
> | | \| | RL[p] vs DP[e] | RL[p] vs BR[e] | \| | DP[p] vs RL[e] | BR[p] vs RL[e] |
> | Grid Map | \| | 1.00 | 1.00 | \| | 11.88 ± 2.39 | 11.67 ± 2.60 |
> | Scotland-Yard Map | \| | 0.99 | 1.00 | \| | 12.57 ± 2.96 | 13.12 ± 3.65 |
> | Downtown Map | \| | 0.99 | 0.99 | \| | 11.83 ± 3.12 | 13.81 ± 5.83 |
> | Times Square, New York | \| | 0.98 | 0.95 | \| | 14.68 ± 3.11 | 16.20 ± 5.16 |
> | Hollywood Walk of Fame, LA | \| | 0.62 | 0.67 | \| | 20.00 ± 4.99 | 20.22 ± 9.60 |
> | Sagrada Familia, Barcelona | \| | 0.66 | 0.76 | \| | 17.89 ± 4.59 | 18.83 ± 6.30 |
> | The Bund, Shanghai | \| | 0.60 | 0.56 | \| | 20.59 ± 5.59 | 21.91 ± 10.58 |
> | Eiffel Tower, Paris | \| | 0.97 | 0.98 | \| | 18.47 ± 6.12 | 19.28 ± 10.48 |
> | Big Ben, London | \| | 0.91 | 0.94 | \| | 21.58 ± 6.38 | 23.07 ± 12.91 |
> | Sydney Opera House, Sydney | \| | 0.74 | 0.80 | \| | 22.37 ± 6.16 | 25.78 ± 16.79 |
>
> **Point 5**
>
> > The framework is evaluated in simplified scenarios, lacking scalability tests on large-scale graphs or complex real-world environments.
>
> Thanks for the comment on scalability. In our training set that contains $152$ graphs, the maximum node number is $295$, and the average node number is $152.24$. Currently, we have conducted additional experiments to further show that our trained policy can directly generalize to large-scale graphs. We increase both map range and discretization accuracy and derive significantly larger graphs for the real-world locations in Table 1. The pursuit success rates for $\mathrm{RL_{p}-DP}$ under the original graphs and the large graphs are shown as follows:
>
> | Map/Location | \| | Original Scale | | | \| | Large Scale | | |
> |---|---|---|---|---|---|---|---|---|
> | | \| | Node Number | \| | Success Rate | \| | Node Number | \| | Success Rate |
> | Downtown Map | \| | 206 | \| | 0.99 | \| | 907 | \| | 0.99 |
> | Times Square, New York | \| | 171 | \| | 0.98 | \| | 768 | \| | 0.89 |
> | Hollywood Walk of Fame, LA | \| | 201 | \| | 0.62 | \| | 892 | \| | 0.71 |
> | Sagrada Familia, Barcelona | \| | 231 | \| | 0.66 | \| | 899 | \| | 0.57 |
> | The Bund, Shanghai | \| | 200 | \| | 0.6 | \| | 952 | \| | 0.54 |
> | Eiffel Tower, Paris | \| | 202 | \| | 0.97 | \| | 616 | \| | 0.82 |
> | Big Ben, London | \| | 192 | \| | 0.91 | \| | 675 | \| | 0.78 |
> | Sydney Opera House, Sydney | \| | 183 | \| | 0.74 | \| | 1074 | \| | 0.83 |
>
> As is shown in the table, the success rates do not suffer from a significant decline under large-scale graphs. In scenarios like Hollywood Walk of Fame and Sydney Opera House, the success rates are even higher because of the structural changes.
>
> Thank you again for providing the insightful comments. We hope our response properly addresses your concerns. We are looking forward to having further discussions with you.
>
> **References**
>
> [1] Tianzhe Chu, Yuexiang Zhai, Jihan Yang, Shengbang Tong, Saining Xie, Dale Schuurmans, Quoc V. Le, Sergey Levine, and Yi Ma. SFT memorizes, RL generalizes: A comparative study of foundation model post-training. In Forty-second International Conference on Machine Learning, 2025.
>
> [2] Pengdeng Li, Shuxin Li, Xinrun Wang, Jakub Cerny, Youzhi Zhang, Stephen McAleer, Hau Chan, and Bo An. Grasper: A generalist pursuer for pursuit-evasion problems. In Proceedings of the 23rd International Conference on Autonomous Agents and Multiagent Systems, pages 1147–1155, 2024.

---

> > ### Comment · Reviewer_BGj6 · 2025-08-05
> >
> > Thank you for your response.
> >
> > I remain unconvinced by the arguments in Point 3. Specifically, the claim that entropy regularization alone can guarantee convergence to the Nash Equilibrium (NE) is problematic. A classic counterexample illustrates this issue: consider a modified Rock-Paper-Scissors (RPS) game where the payoffs for both players are doubled if at least one player chooses "Paper". In this scenario, the NE shifts from the uniform distribution (1/3, 1/3, 1/3) to a non-uniform strategy profile—specifically, (Scissors: 2/5, Rock: 2/5, Paper: 1/5).
> >
> > However, entropy-regularized algorithms like SAC, which inherently favor maximizing policy entropy, would still converge to the uniform (1/3, 1/3, 1/3) distribution in this case. Thus, if an algorithm can perform arbitrarily poorly on each single map, it becomes difficult to guarantee the unexploitability of the results trained on a diverse set. More analysis are needed to support the authors' claims.
> >
> > Regarding point 5, the observation that the policy performs even better on the larger map is quite surprising. Could you elaborate on the difference between the large and the small maps? Are the policies finetuned on the larger map?

---

> > > ### Author Response · Authors · 2025-08-06
> > >
> > > Thank you for the questions. Here we provide further explanations and analysis for Point 3 and Point 5.
> > >
> > > **Point 3**
> > >
> > > Thanks for further commenting on Point 3. We agree that the SAC loss always favors distributions with a large entropy and does not lead to the NE $(2/5,2/5,1/5)$ in the RPS variation. We also agree that classic equilibrium learning algorithms, which must guarantee convergence to the NE given the game dynamics, cannot be constructed by simply optimizing against a fixed NE. Our original response is somewhat misleading since it appears to emphasize the capability of SAC. For the RPS variation, equilibrium guidance is necessary for EPG if our purpose is to guarantee the convergence to the NE in this single game.
> > >
> > > When it comes to cross-graph/dynamics training, however, it is underexamined whether pure RL can learn a meaningful high-level policy against equilibrium adversaries under SAC loss, which encourages explorations during training. There is certain empirical evidence to show that it is more likely to happen when the graph set is diverse enough. The orange lines in Figure 3 show that pure RL struggles at first but eventually finds a reasonable cross-graph policy under the training corpus with 152 graphs. Actually, when trained on a set with only 10 graphs (our preliminary experiment), both pure RL and equilibrium-guided RL fail to learn a policy with a desirable performance on the testing graphs. That is to say, training under 1 graph, 10 graphs, and 100 graphs can have significantly different results.
> > >
> > > Here is our understanding that may help explain this phenomenon arising from cross-graph training. For dynamic games like PEGs, they can have more complicated structures in comparison with static games like RPS. For example, [1] shows that strategies in many real-world games can have different levels of transitive strength, with NE being the strongest. In a single PEG, RL against NE has the benefit of excluding the policies that are significantly weaker. Cross-graph training is similar to finding the joint part of the remaining strategies and abstracting them into a high-level policy. When the training graphs are diverse enough, the divisions on the policy space through RL against single-graph NE can be different from each other. Let us imagine that a half space is left after each division (exclusion) and the divisions from different graphs are mutually orthogonal. Then, the high-level policy will be improved at an exponential level in the number of graphs. This rough analysis could account for the sharp distinction between learning under a limited number of graphs and a diverse set of graphs.
> > >
> > > **Point 5**
> > >
> > > Thanks for the question on Point 5. The larger maps are generated by doubling the discretization accuracy and increasing the map range. For Sydney Opera House, we actually double the map range to create a graph with at least $1000$ nodes. We do not fine-tune our policy on the larger map. We only adjust the termination function to keep the difficulty of the real-world problem close to the original one. Specifically, one pursuer succeeds when its distance to the evader is no greater than $1$ in the original graph and when the distance is no greater than $2$ in the larger graph. Since the discretization accuracy doubles, the two conditions correspond to roughly the same real-world pursuit condition despite some local structural changes. Under the new graphs with the new condition, we rerun DP to generate the optimized policies correspondingly.
> > >
> > > In our response to Reviewer LAVP (Question 2), we provide a rough analysis about how graph structures affect pursuit success rates. The analysis could similarly account for the higher success rate of Sydney Opera House. When the map range increases, the number of "branches" that have relatively poor connectivity with other nodes actually increases at this real-world location. Since our initial positions are randomized over the entire graph, the success rate can be higher consequently, as the evaders that are initially close to these branches would be easier for pursuit.
> > >
> > > We hope this response better clarifies Point 3 and Point 5. We are looking forward to your further comments. If you still hold questions, we are definitely willing to make further clarifications.
> > >
> > > **Reference**
> > >
> > > [1] Wojciech M Czarnecki, Gauthier Gidel, Brendan Tracey, Karl Tuyls, Shayegan Omidshafiei, David Balduzzi, and Max Jaderberg. Real world games look like spinning tops. Advances in Neural Information Processing Systems, 33:17443–17454, 2020.

---

> > > > ### Comment · Reviewer_BGj6 · 2025-08-06
> > > >
> > > > Thank you for your response. I will raise my score to 4, as the authors has addressed my concerns.

---

> > > > > ### Author Response · Authors · 2025-08-06
> > > > >
> > > > > We are delighted that our response and additional results help to address your concerns. Thank you again for your valuable comments.

---

### Author Response · Authors · 2025-08-05
**Response to All Reviewers**

Dear reviewers,

Thank you for reviewing this paper. We greatly appreciate your effort in reading the paper and providing valuable comments. Here we summarize the additional results that we have included in our revised paper according to the suggestions from all of you.

**Result against approximate best response (BR) in comparison with DP:**

| Map/Location | \| | Pursuit Success Rate | | \| | Evasion Timestep | |
|---|---|---|---|---|---|---|
| | \| | RL[p] vs DP[e] | RL[p] vs BR[e] | \| | DP[p] vs RL[e] | BR[p] vs RL[e] |
| Grid Map | \| | 1.00 | 1.00 | \| | 11.88 ± 2.39 | 11.67 ± 2.60 |
| Scotland-Yard Map | \| | 0.99 | 1.00 | \| | 12.57 ± 2.96 | 13.12 ± 3.65 |
| Downtown Map | \| | 0.99 | 0.99 | \| | 11.83 ± 3.12 | 13.81 ± 5.83 |
| Times Square, New York | \| | 0.98 | 0.95 | \| | 14.68 ± 3.11 | 16.20 ± 5.16 |
| Hollywood Walk of Fame, LA | \| | 0.62 | 0.67 | \| | 20.00 ± 4.99 | 20.22 ± 9.60 |
| Sagrada Familia, Barcelona | \| | 0.66 | 0.76 | \| | 17.89 ± 4.59 | 18.83 ± 6.30 |
| The Bund, Shanghai | \| | 0.60 | 0.56 | \| | 20.59 ± 5.59 | 21.91 ± 10.58 |
| Eiffel Tower, Paris | \| | 0.97 | 0.98 | \| | 18.47 ± 6.12 | 19.28 ± 10.48 |
| Big Ben, London | \| | 0.91 | 0.94 | \| | 21.58 ± 6.38 | 23.07 ± 12.91 |
| Sydney Opera House, Sydney | \| | 0.74 | 0.80 | \| | 22.37 ± 6.16 | 25.78 ± 16.79 |

**Result against PSRO:**

| Map/Location | \| | Timestep (Success Rate) | | | |
|---|---|---|---|---|---|
| | \| | RL[p] vs PSRO[e] | \| | PSRO[p] vs RL[e] | |
| Grid Map | \| | 14.04 ± 6.88 (1.00) | \| | 11.87 ± 2.39 (1.00) | |
| Scotland-Yard Map | \| | 17.45 ± 9.17 (1.00) | \| | 25.48 ± 16.80 (1.00) | |
| Downtown Map | \| | 14.14 ± 7.37 (1.00) | \| | 38.39 ± 30.50 (0.97) | |
| Times Square, New York | \| | 17.72 ± 7.86 (1.00) | \| | 37.87 ± 29.94 (0.97) | |
| Hollywood Walk of Fame, LA | \| | 30.93 ± 17.67 (1.00) | \| | 60.71 ± 39.21 (0.86) | |
| Sagrada Familia, Barcelona | \| | 26.06 ± 12.43 (1.00) | \| | 39.50 ± 28.25 (0.97) | |
| The Bund, Shanghai | \| | 24.76 ± 13.01 (1.00) | \| | 34.97 ± 24.75 (0.99) | |
| Eiffel Tower, Paris | \| | 19.10 ± 10.69 (1.00) | \| | 25.24 ± 14.07 (1.00) | |
| Big Ben, London | \| | 23.70 ± 13.00 (1.00) | \| | 53.67 ± 36.41 (0.90) | |
| Sydney Opera House, Sydney | \| | 26.14 ± 12.81 (1.00) | \| | 67.23 ± 43.37 (0.77) | |

**Scalability test:**

| Map/Location | \| | Original Scale | | | \| | Large Scale | | |
|---|---|---|---|---|---|---|---|---|
| | \| | Node Number | \| | Success Rate | \| | Node Number | \| | Success Rate |
| Downtown Map | \| | 206 | \| | 0.99 | \| | 907 | \| | 0.99 |
| Times Square, New York | \| | 171 | \| | 0.98 | \| | 768 | \| | 0.89 |
| Hollywood Walk of Fame, LA | \| | 201 | \| | 0.62 | \| | 892 | \| | 0.71 |
| Sagrada Familia, Barcelona | \| | 231 | \| | 0.66 | \| | 899 | \| | 0.57 |
| The Bund, Shanghai | \| | 200 | \| | 0.6 | \| | 952 | \| | 0.54 |
| Eiffel Tower, Paris | \| | 202 | \| | 0.97 | \| | 616 | \| | 0.82 |
| Big Ben, London | \| | 192 | \| | 0.91 | \| | 675 | \| | 0.78 |
| Sydney Opera House, Sydney | \| | 183 | \| | 0.74 | \| | 1074 | \| | 0.83 |

Besides, we have also updated the contents in Table 1 and Table 4 with standard deviations and RL vs RL. The explanations for all of the additional results are included in our responses to each of you and serve to address the weaknesses.

In addition, here we further clarify several basic points that may help address potential misunderstandings about this paper:

1. The only execution time cost of the EPG policy is from the $O(|V|^3)$ Floyd algorithm (which is finished within 1 second even under 1000-node graphs) that computes the shortest path distances between vertices. Besides, this time cost is necessary only when the change of graph structure is detected. Otherwise, we can simply reuse the old shortest-path-distance table. Equilibrium oracles are only required for the graphs in the training corpus.

2. The diversity of the training graphs is important for robust zero-shot generalization of EPG. Our preliminary tests show that when the training set only contains 10 graphs rather than 152 graphs, the RL pursuers can struggle to capture the DP evader during testing just like SPS pursuers. Further increasing graph diversity can further improve our current zero-shot performance.

3. Cross-graph training against equilibrium adversaries is the core of this paper. Without this idea, zero-shot generalization (especially in no-exit scenarios) is underexplored in current PEG research. Besides, equilibrium adversary is different from equilibrium guidance. The latter is a technique that improves the training efficiency by applying a KL regularization to the original policy loss based on the reference policy (rather than the opponent policy).

We will be happy if our responses and modifications can address your concerns and improve your appreciation of this work. If you find something unclear, please feel free to point it out. We are more than delighted to have further discussions and improve this paper.

---

### Note · Authors · 2025-08-15

We would like to thank all reviewers again for the time spared in reviewing and discussing our paper. We are happy to see that our responses address the major concerns during the discussion period. The valuable comments helped us to make further modifications that better clarify the essence and applicability of the proposed methods, which include the Equilibrium Policy Generalization (EPG) framework that facilitates robust zero-shot generalization in adversarial games and the specific algorithms (DP and heuristic) that serve to preprocess exact or approximate equilibrium policies in no-exit or multi-exit pursuit-evasion games (PEGs). We hope the discussion contents (some of which have already been included in our current paper) can provide a better understanding of this work and also facilitate potential RL research concerning multi-agent adversarial games and zero-shot generalization across environment dynamics.

---

### Decision · Program_Chairs · 2025-09-17

**Decision:**

Accept (poster)

**Comment:**

This paper proposes using reinforcement learning to learn pursuit-evasion game policies that generalize across different graph environments. The authors simulate the adversary's policy using a dynamic programming algorithm for no-exit Markov PEGs and a heuristic for multi-exit cases. They then train a soft-actor-critic agent against this equilibrium adversary. Experiments on real-world graphs demonstrate zero-shot generalization of the learned policy. Reviewers stated many strengths of the paper, such as that EPG’s focus on zero-shot generalization via equilibrium guidance is innovative and practical, the core idea is intuitive and the motivation for equilibrium-guided reinforcement learning is well explained, and experiments on diverse real-world maps demonstrate the method's strong zero-shot performance. The authors did a great job during the rebuttal phase and addressed many of the concerns raised by the reviewers, which led to a significant increase in the rating. Although some reviewers still have reservations about applying the proposed EPG policies to more complex scenarios with larger agent scales, I believe the work deserves a chance to be evaluated by a much broader readership in the future, and the results presented in the paper could benefit researchers in related fields. Therefore, I recommend acceptance.